# *BAPFL*: EXPLORING BACKDOOR ATTACKS AGAINST PROTOTYPE-BASED FEDERATED LEARNING

## ABSTRACT

Prototype-based federated learning (PFL) has emerged as a promising paradigm to address data heterogeneity problems in federated learning, as it leverages mean feature vectors as prototypes to enhance model generalization. However, its robustness against backdoor attacks remains largely unexplored. In this paper, we identify that PFL is inherently resistant to existing backdoor attacks due to its unique prototype learning mechanism and local data heterogeneity. To further explore the security of PFL, we propose BAPFL, the first backdoor attack method specifically designed for PFL frameworks. BAPFL integrates a prototype poisoning strategy with a trigger optimization mechanism. The prototype poisoning strategy manipulates the trajectories of global prototypes to mislead the prototype training of benign clients, pushing their local prototypes of clean samples away from the prototypes of trigger-embedded samples. Meanwhile, the trigger optimization mechanism learns a unique and stealthy trigger for each potential target label, and guides the prototypes of trigger-embedded samples to align closely with the global prototype of the target label. Experimental results across multiple datasets and PFL variants demonstrate that BAPFL achieves a 33%-75% improvement in attack success rate compared to traditional backdoor attacks, while preserving main task accuracy. These results highlight the effectiveness, stealthiness, and adaptability of BAPFL in PFL.

## 1 INTRODUCTION

Federated learning (FL) is a distributed machine learning paradigm that enables multiple clients to collaboratively train a global model without sharing their private data, thus preserving data privacy. Due to this advantage, FL has been widely applied in various real-world scenarios, such as personalized recommendation (Zhang et al., 2024), autonomous driving (Li et al., 2022), and smart healthcare (Liu et al., 2022). In such practical applications, however, clients usually gather data from diverse sources, resulting in significant data heterogeneity. This data heterogeneity makes it challenging for a unified global model to achieve high performance on all clients. To address this challenge, many studies have focused on heterogeneous FL (Yan et al., 2025; Tang et al., 2024; Zhou et al., 2024; Tan et al., 2022a). Among these approaches, prototype-based federated learning (PFL) (Tan et al., 2022a) has shown great promise due to its ability to learn high-quality personalized models for clients with minimal communication overhead.

Unlike vanilla FL methods that aggregate full model parameters across clients, PFL (Tan et al., 2022a) exchanges *class prototypes*, i.e., the average feature vectors of samples within the same class, to train models for clients. Typically, each client periodically updates its local prototypes and model by minimizing classification loss and aligning local prototypes with the global prototypes. The server then averages these local prototypes by class to form new global prototypes. Compared to FL, PFL significantly reduces communication overhead and improves model generalization under heterogeneous data (Tan et al., 2022b; 2025). With ongoing innovations in prototype representation (Tan et al., 2022b; Huang et al., 2023b; Fu et al., 2025b), optimization objective (Wang et al., 2024), and robust aggregation (Tan et al., 2025; Yan et al., 2024), PFL is expected to play a key role in real-world heterogeneous FL systems.

Despite its potential, the security of PFL remains underexplored. This research gap creates a critical blind spot: as illustrated in Figure 1(a), attackers can exploit the prototype-sharing mechanism of

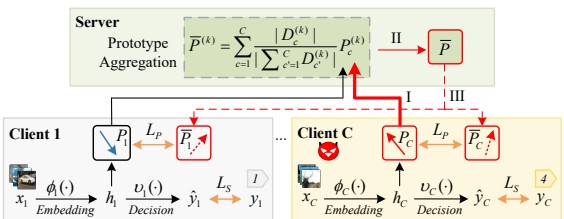 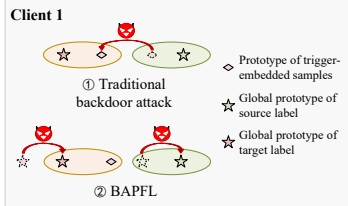

(a) PFL workflow and potential attack process      (b) Traditional attack vs. BAPFL

Figure 1: The left subgraph illustrates the PFL system and its potential attack threat. The attacker manipulates client $C$ to upload poisoned prototypes $P_C$ (step I). $P_C$ deviates from the benign prototypes like $P_1$, thus it poisons the global prototypes $\bar{P}$ (step II) and misleads the local training of benign models (step III). The right subgraph compares the strategies of the traditional backdoor attack (①) and our BAPFL attack (②).

PFL by manipulating some clients to upload poisoned prototypes (step I). These prototypes deviate from benign prototypes uploaded by benign clients and cause the aggregated prototypes to drift from the correct direction (step II), thereby misleading the local training of multiple benign clients (step III). Affected by such attacks, the PFL system may cause serious consequences in security-critical applications such as medical diagnosis and financial decision-making.

Among various federated attack strategies, backdoor attacks pose a particularly insidious and dangerous threat (Feng et al., 2025). These attacks inject poisoned samples with specific triggers into the training data to manipulate the model's predictions. In this paper, we investigate the susceptibility of PFL frameworks to backdoor attacks. We first explore whether the PFL approach is still vulnerable to existing backdoor attacks. We observe that PFL exhibits strong robustness against existing backdoor attacks (see Section 3 for details). We attribute this robustness to two key factors: 1) The limited influence of poisoned prototypes. Even if the global prototypes are contaminated by poisoned prototypes, they only affect the embedding layer of benign models. While the unaffected decision layer of the benign model obstructs the attack effectiveness. 2) Data heterogeneity of clients. Some clients may lack the training samples of the target label. Thus, their decision layer does not learn parameters for the target label. This inherently breaks the *trigger-target label* mapping and significantly reduces the attack success rate (ASR).

These factors motivate us to rethink backdoor attack strategies for PFL. As illustrated in Figure 1(b), while traditional backdoor attack can directly manipulate the trigger-embedded samples' classification in traditional FL by sharing full model parameters, we must strategically manipulate the global prototype to mislead the trigger-embedded samples' classification in an indirect manner. According to this analysis, we propose BAPFL, a novel backdoor attack method designed for PFL. BAPFL effectively attacks PFL systems from the perspective of dual-direction prototype optimization. Specifically, BAPFL comprises two components: 1) A prototype poisoning strategy (PPS) that leverages poisoned prototypes to manipulate the global prototype away from the prototypes of trigger-embedded samples (termed trigger prototypes), thereby guiding benign prototypes away from these trigger prototypes. 2) A trigger optimization mechanism (TOM) that ensures the attack's effectiveness across heterogeneous clients. It learns stealthy triggers for target labels, and optimizes the trigger prototypes to closely align with the global prototype of the target label. These two modules jointly enhance the effectiveness of BAPFL, achieving high ASR and main task accuracy (ACC) across diverse PFL frameworks. Our main contributions are summarized as follows.

- This study delves into the security domain of PFL, and reveals that PFL exhibits strong resistance to conventional backdoor attacks. We identify two key factors behind this resistance: the limited influence of poisoned prototypes and data heterogeneity of clients.

- We propose a novel backdoor attack method for PFL, called BAPFL, which combines PPS and TOM. PPS pushes benign prototypes away from trigger prototypes by manipulating global prototype aggregation, while TOM pulls trigger prototypes closer to the global prototypes of target labels by learning diverse stealthy triggers. This dual-direction prototype optimization design enhances the effectiveness of BAPFL.

- We quantitatively evaluate the performance of BAPFL in PFL based on representative datasets including MNIST (LeCun et al., 1998), FEMNIST (Caldas et al., 2019), and CIFAR-10 (Krizhevsky, 2009). Results show that BAPFL achieves a 35%-75% increase in ASR while maintaining ACC. We also integrate BAPFL into different PFL frameworks and heterogeneous settings, and the results highlight its broad adaptability.

## 2 RELATED WORK

To address the challenges posed by data heterogeneity in FL, existing solutions can be categorized into model-based and data-based methods. Model-based methods aim to enhance the final model's ability to adapt to the diverse data distributions of clients. For instance, FedProx (Li et al., 2020) introduces a proximal regularization term to restrict model divergence. Personalized FL approaches (Zhang et al., 2023d; Lyu et al., 2024; Fan et al., 2025; Ye et al., 2024) enable each client to maintain individualized models to better fit their local data. EAFL (Zhou et al., 2024) and HCFL (Guo et al., 2025) partition clients into groups with similar data distributions and train separate global models per cluster. However, these methods incur high communication overhead. Data-based methods are a more communication-efficient alternative, focusing on learning shared representations across clients. For example, Fed2KD (Wen et al., 2023) shares knowledge across clients to boost the model accuracy. GPFL (Zhang et al., 2023c) and FedCR (Zhang et al., 2023b) extract global and personalized features/representations to enhance model generalization. PFL (Tan et al., 2022a; Mu et al., 2023; Tan et al., 2022b; Jiang et al., 2025; Fu et al., 2025a; Tan et al., 2025) leverages class prototypes to align local and global semantics. FPL (Huang et al., 2023a) and FedPLVM (Wang et al., 2024) further build hierarchical and unbiased prototypes for better learning performance. In this paper, we focus on PFL and explore its security threats.

Backdoor attacks have proven effective in vanilla FL, and are typically categorized into data poisoning attacks (Feng et al., 2025) and model poisoning attacks (Bagdasaryan et al., 2020; Xie et al., 2020; Liu et al., 2024). Data poisoning attacks inject trigger-embedded samples into local datasets to poison local models. In contrast, model poisoning attacks directly manipulate model updates for stronger attack effectiveness. Representative methods include model replacement (MR) (Bagdasaryan et al., 2020), which scales malicious updates to pollute the aggregated model but suffers from dilution by subsequent benign updates. To enhance the persistence of the attack, Bad-PFL (Fan et al., 2025) employs features from natural data as the trigger, while distributed backdoor attack (DBA) (Xie et al., 2020) distributes trigger fragments across clients. Full combination backdoor attack (FCBA) (Liu et al., 2024) further creates diverse trigger variants to increase the ASR. Additionally, BapFL (Ye et al., 2024) poisons the encoder layers and simulate classifiers to implant effective triggers, PFedBA (Lyu et al., 2024) and 3DFed (Li et al., 2023) incorporate anomaly-aware loss functions to improve attack stealthiness. Chameleon (Dai & Li, 2023) adapts the trigger pattern to the evolving global model to maintain attack effectiveness under aggregation perturbations, while A3FL (Zhang et al., 2023a) introduces adaptive gradient manipulation to preserve attack persistence against common FL defenses.

To defend against backdoor attacks, existing solutions typically introduce defense strategies to identify and eliminate abnormal models in FL. Multi-Krum (Blanchard et al., 2017) selects the most reliable client updates by evaluating the distance between updates and choosing those that are least affected by outliers. Median (Zhang et al., 2023e) aggregates model updates by selecting the median across each parameter dimension, resisting interference from extreme values. Sign (Guo et al., 2023) processes the sign of model updates to enhance FL robustness against malicious updates. To further resist backdoor attacks under Non-IID data settings, FLAME (Nguyen et al., 2022) combines differential privacy, norm clipping, and weight clustering to filter out potential malicious updates. Additionally, Deepsight (Rieger et al., 2022) introduces a deep model inspection framework that analyzes local updates to identify potential backdoor threats.

While significant progress has been made in designing and resisting backdoor attacks in FL, little attention has been paid to the threat of backdoor attacks in PFL. In this paper, we fill this gap and propose an effective backdoor attack specifically designed for PFL.

## 3 MODELING AND ANALYSIS

### 3.1 FL VERSUS PFL

Consider a FL system with $C$ clients (denoted as $\mathcal{C} = \{1, \ldots, c, \ldots, C\}$) and a central server (Huang et al., 2024). Each client $c$ holds its dataset $\mathcal{D}_c = \{(x_c^i, y_c^i)\}_{i=1}^{|\mathcal{D}_c|}$. The local training objective is:

$$\arg\min_\theta \ \mathcal{L}_S = \frac{1}{|\mathcal{D}_c|} \sum_{i=1}^{|\mathcal{D}_c|} \ell(f_\theta(x_c^i), y_c^i), \tag{1}$$

where $\ell(\cdot, \cdot)$ denotes the loss of supervised learning, and $f_\theta$ is the model parameterized by $\theta$. In each round, clients send model updates to the server for aggregation. However, under heterogeneous data, the aggregated model may perform poorly on some clients.

PFL (Tan et al., 2022a) mitigates this issue by exchanging local prototypes, i.e., mean feature vectors, instead of model updates to enhance model generalization. As illustrated in Figure 1(a), each client shares a common feature extractor $\phi(\cdot)$, and computes the local class prototype for class $k$ as:

$$P_c^{(k)} = \frac{1}{|\mathcal{D}_c^{(k)}|} \sum_{(x_c^i, y_c^i) \in \mathcal{D}_c^{(k)}} \phi(x_c^i), \tag{2}$$

where $\mathcal{D}_c^{(k)} = \{(x_c^i, y_c^i) \in \mathcal{D}_c \mid y_c^i = k\}$. Then, the server aggregates local prototypes via:

$$\bar{P}^{(k)} = \sum_{c=1}^C \frac{|\mathcal{D}_c^{(k)}|}{\sum_{c'=1}^C |\mathcal{D}_{c'}^{(k)}|} P_c^{(k)}. \tag{3}$$

Subsequently, client $c$ optimizes its local model using its private data and the global prototypes $\bar{P} = \{\bar{P}^{(k)}\}_{k=1,2,\ldots}$ by minimizing a combined loss $\mathcal{L}$, which includes the supervised loss $\mathcal{L}_S$ and a prototype regularization term $\mathcal{L}_P$, i.e.,

$$\mathcal{L} = \mathcal{L}_S + \lambda \cdot \mathcal{L}_P = \frac{1}{|\mathcal{D}_c|} \sum_{i=1}^{|\mathcal{D}_c|} [\ell(f_\theta(x_c^i), y_c^i) + \lambda \cdot \|\phi(x_c^i) - \bar{P}^{(y_c^i)}\|_2], \tag{4}$$

where $\lambda$ is the coefficient that controls the trade-off between $\mathcal{L}_S$ and $\mathcal{L}_P$.

### 3.2 THREAT MODEL

**Adversary's Goal**. Similar to previous backdoor attacks (Xie et al., 2020; Feng et al., 2025), we consider an adversary that can control multiple compromised clients to upload poisoned prototypes after local training. Its goal is to contaminate benign clients' models such that they misclassify trigger-embedded samples as the target label, while maintaining high test accuracy on clean samples. The adversary further aims for the backdoor to be stealthy and persistent, avoiding detection and removal throughout training.

**Adversary's Knowledge and Capability**. The adversary fully controls the compromised clients, along with their data, training process, and the received global prototypes. However, the adversary cannot control the server and the benign clients. That is, the adversary cannot modify the aggregation rules or interfere with the training process of benign clients.

### 3.3 CHALLENGES OF BACKDOOR ATTACKS IN PFL

We consider a standard federated backdoor attack, where each compromised client $c^*$ poisons other benign models by inserting a backdoor task into its local model training. During training, client $c^*$ minimizes the following loss:

$$\mathcal{L}_S^* = (1-\alpha) \cdot \frac{1}{|\mathcal{D}_{c^*}|} \sum_{i=1}^{|\mathcal{D}_{c^*}|} \ell(f_\theta(x_{c^*}^i), y_{c^*}^i) + \alpha \cdot \frac{1}{m_{c^*}} \sum_{j=1}^{m_{c^*}} \ell(f_\theta(T(x_{c^*}^j)), y_t), \tag{5}$$

where $T(\cdot)$ is the trigger function that injects a trigger to the training samples and assigns them a target label $y_t$, $m_{c^*}$ is the number of poisoned samples, and $\alpha$ is the poisoning ratio that controls the importance of the backdoor task relative to the main task. In PFL, however, we observe this standard backdoor attack consistently yields low ASR. We identify two key factors for this failure:

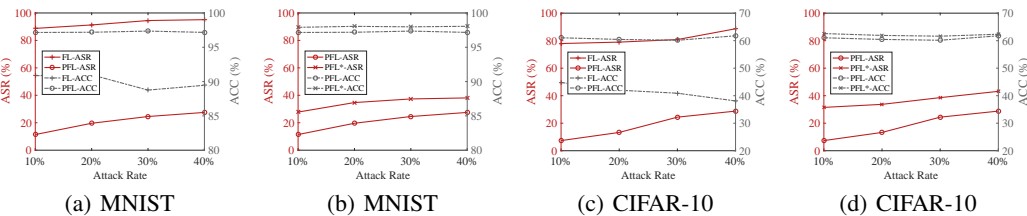

Figure 2: The ASR and ACC of FL, PFL and PFL* under backdoor attacks.

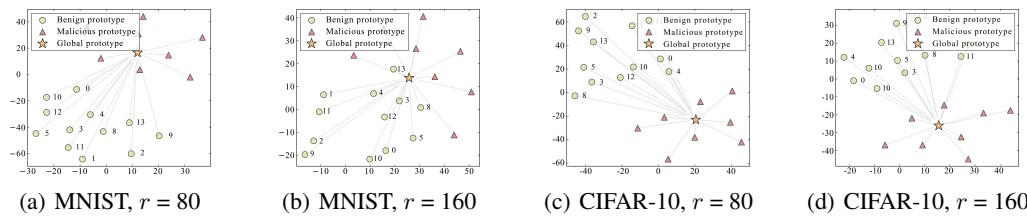

(a) MNIST, $r = 80$     (b) MNIST, $r = 160$     (c) CIFAR-10, $r = 80$     (d) CIFAR-10, $r = 160$

Figure 3: t-SNE visualization of benign and poisoned prototypes at different training rounds $r$.

**The limited influence of poisoned prototypes**. In FL, the attacker can effectively embed backdoor effect by poisoning all benign model parameters. In PFL, however, the attacker can only affect the embedding layer of benign models via $\mathcal{L}_P$. Its attack effectiveness is obstructed by the unaffected decision layer of the benign model. To validate this, we assess the performance of the backdoor attack against the standard PFL and FL frameworks. Detailed experimental settings are provided in Appendix B. The results are shown in Figure 2(a) and Figure 2(c). We find that the ASR in FL setting remains above 70%, while the ASR in PFL reaches only around 10%-20%. This indicates that global prototypes exert limited influence on the decision-making of benign clients' models, thereby obstructing the backdoor propagation path.

**Data heterogeneity of clients**. In PFL, some clients do not contain the training samples of the target label $y_t$. Thus their models lack classifier parameters for $y_t$, inherently avoiding the mapping from trigger to $y_t$. We confirm this via an ablation study, in which we inject the training samples of $y_t$ into all clients under PFL (denoted as "PFL*") and compare it with the original PFL. As shown in Figure 2(b) and Figure 2(d), the attacker in PFL* achieves higher ASR across all attack rates.

The above challenges motivate us to rethink backdoor attack strategies in PFL. We first examine the distribution changes of global and benign prototypes at the 80-th and 160-th training rounds (TR) in PFL under backdoor attacks, and the results are shown in Figure 3. We observe that, as the number of training round $r$ increases, benign prototypes gradually converge toward the manipulated global prototype. This motivates us to develop a novel attack strategy: ***by manipulating the global prototype away from the trigger prototype, the attacker may indirectly push benign prototypes away from the trigger prototypes, thereby increasing the probability of misclassifying trigger-embedded samples***. To further classify the trigger-embedded samples into the target label, the trigger prototypes can be optimized toward the global prototype of the target label.

## 4 PROPOSED BAPFL: BACKDOOR ATTACK AGAINST PROTOTYPE-BASED FEDERATED LEARNING

### 4.1 OVERVIEW

Based on the analysis in Section 3, we propose a novel backdoor attack method BAPFL, which exploits the dual-direction prototype optimization mechanism to indirectly propagate backdoor behavior across diverse PFL frameworks (Tan et al., 2022a;b; 2025). As illustrated in Figure 4 and Appendix C. BAPFL integrates two components: *prototype poisoning strategy* and *trigger optimization mechanism*. Each malicious client $c^*$ executes PPS and TOM to generate poisoned prototypes $P_{c^*}$

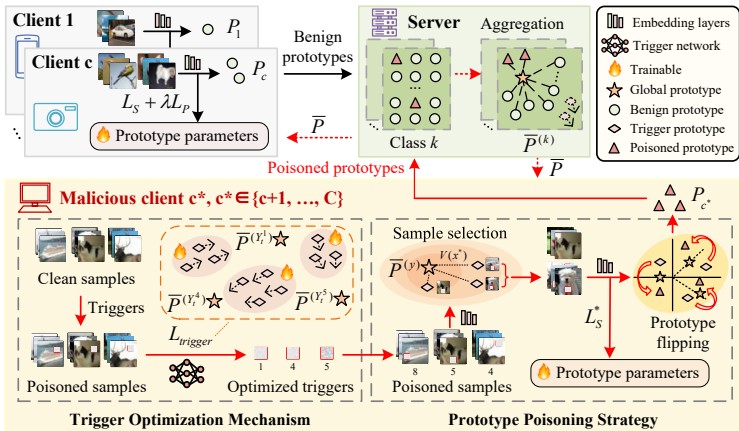

Figure 4: Overview of BAPFL in PFL.

and optimize label-specific triggers, respectively. Specifically, the poisoned prototypes of PPS deliberately bias the global prototypes away from the trigger prototypes. This manipulation indirectly influences the prototype learning of benign clients, pushing their benign prototypes to diverge from the trigger prototypes. As this discrepancy increases, trigger-embedded samples are more likely to be misclassified by benign models. Meanwhile, TOM expands the target label space and optimizes triggers for each target label. This increases the probability that the target labels overlap with the local label space of benign clients, thereby enabling the benign models to inadvertently activate the *trigger-target label* mapping. Further theoretical analysis of BAPFL is provided in Appendix E.

## 4.2 PROTOTYPE POISONING STRATEGY (PPS)

PPS includes two steps: sample selection and prototype flipping. The former identifies the most valuable trigger-embedded samples. The prototypes of these selected samples serve as the basis for the latter, which constructs the accurate poisoned prototypes in a deliberately opposite direction, thereby manipulating the global prototypes away from the trigger prototypes.

**Sample Selection Strategy**. Malicious clients first compute the attack value of each trigger-embedded sample $x^*$ by computing the Euclidean distance between its prototype $\phi(x^*)$ and the global prototype $\bar{P}^{(y)}$, where $y$ is the ground-truth label of the clean sample $x$ related to $x^*$. That is,

$$V(x^*) = \|\phi(x^*) - \bar{P}^{(y)}\|_2, \tag{6}$$

where a larger distance implies higher attack value. Compared with other alternative measures such as cosine similarity or projection, we employ Euclidean distance because PFL is inherently optimized based on the $L_2$ norm. This alignment enables the Euclidean distance to better reflect both directional and magnitude shifts introduced by triggers (see Appendix G). The top-$K$ samples with the highest attack values are selected for training local model and constructing poisoned prototypes.

**Prototype Flipping Strategy**. To mislead the global prototype, malicious clients construct poisoned prototypes and upload them to the server. Specifically, malicious client $c^*$ first computes the class-wise trigger prototype $P_{tr}^{(k)}$ from the selected samples. Then, $c^*$ computes the projection of $P_{tr}^{(k)}$ onto the corresponding global prototype $\bar{P}^{(k)}$. Finally, $c^*$ constructs the poisoned prototype $P_{c^*}^{(k)}$ by performing a symmetrical flip of $P_{tr}^{(k)}$ with respect to this projection, i.e.,

$$P_{c^*}^{(k)} = 2 \cdot P_{proj} - P_{tr}^{(k)}, \tag{7}$$

where $P_{proj} = \frac{\bar{P}^{(k)} \cdot P_{tr}^{(k)}}{\bar{P}^{(k)} \cdot \bar{P}^{(k)}} \cdot \bar{P}^{(k)}$ denotes the projection of $P_{tr}^{(k)}$ onto $\bar{P}^{(k)}$. Compared with other flipping strategies such as origin-based or global prototype-based symmetry, our proposed prototype flipping strategy achieves finer control over both the direction and the norm of poisoned prototypes (see Appendix H). This ensures effective and stealthy attack for benign clients' prototype learning.

### 4.3 TRIGGER OPTIMIZATION MECHANISM (TOM)

TOM includes two steps: trigger optimization and trigger training. The former designates a specific trigger to each target label, while the latter trains triggers for optimal effectiveness and stealthiness.

**Trigger Optimization Strategy**. To enhance attack effectiveness across benign clients with heterogeneous data, we expand the attack's target label space from a single label $y_t$ to a label set $Y_t$ that encompasses all local labels of benign clients. For each target label $y_t \in Y_t$, a specific trigger $(\delta_{y_t}, M_{y_t})$ is learned. This design enables BAPFL to perform personalized backdoor attacks on benign clients. Specifically, if the local label space of benign client $c$ contains the target label $y_t$, BAPFL can activate backdoor behaviors of $c$'s local model, enabling it to classify the samples embedded with the trigger $(\delta_{y_t}, M_{y_t})$ as $y_t$.

**Trigger Training Strategy**. For each target label $y_t$, we learn a dedicated trigger pattern $\delta_{y_t}$ and a corresponding mask $M_{y_t}$, forming a trigger function $T_{y_t}(x) = (1 - M_{y_t}) \odot x + M_{y_t} \odot \delta_{y_t}$. The optimization objective of each trigger aims to simultaneously: 1) minimize classification loss of $T_{y_t}(x)$ to $y_t$, 2) align the prototype of $T_{y_t}(x)$ with the global prototype $\bar{P}^{(y_t)}$, and 3) ensure visual imperceptibility of the trigger. The corresponding loss function is formulated as:

$$\mathcal{L}_{trigger} = \underbrace{\mathcal{L}_S(f_\theta(T_{y_t}(x)), y_t)}_{\text{target classification loss}} + \lambda_1 \cdot \underbrace{\|\phi(T_{y_t}(x)) - \bar{P}^{(y_t)}\|_2}_{\text{prototype alignment}} + \lambda_2 \cdot \underbrace{\|M_{y_t}\|_1 + \lambda_3 \cdot \|\delta_{y_t}\|_2}_{\text{stealthiness loss}}, \quad (8)$$

where $\lambda_1$, $\lambda_2$, and $\lambda_3$ are hyperparameters balancing effectiveness and stealthiness.

## 5 PERFORMANCE EVALUATION

### 5.1 EXPERIMENT SETUP

**Datasets and Models**. Our experiments are conducted on four datasets: MNIST (LeCun et al., 1998), FEMNIST (Caldas et al., 2019), CIFAR-10 (Krizhevsky, 2009) and CIFAR-100 (Krizhevsky, 2009), which are benchmark datasets for image classification. The dataset and model details are provided in Appendix B.

**Training Setting**. We choose FedProto (Tan et al., 2022a) as the basic PFL framework. We set 20 clients and 200 training rounds. In each training round, each client performs 1 local epoch with a local batch size of 4, and the learning rate is set to 0.01. We assume that all clients perform learning tasks with heterogeneous statistical distributions. Specifically, each client is assigned a $p$-way $q$-shot classification task, where $p$ and $q$ denote the maximum number of local classes and samples per class, respectively. We also use a Dirichlet distribution with parameter $\beta$ for data sampling. By default, we set $p = 5$, $q = 100$, and $\beta = 0.5$. Additionally, we set $\alpha = 0.75, \lambda = 1, \lambda_1 = 0.1, \lambda_2 = 0.01$ and $\lambda_3 = 0.001$ (see Appendix I for further analyses). All experiments are repeated 10 times, and the average results $\pm$ standard deviation are reported.

**Attack Setup**. We simulate a backdoor attack scenario with attack rates (AR) of 10%, 20%, 30%, and 40%, where AR denotes the proportion of malicious clients controlled by the attacker. For each ground-truth label $y$, we randomly assign a different label (i.e., $y' \neq y$) as its target label. The compromised clients train triggers for their respective target labels over 50 local rounds and embed them into their local data to construct poisoned prototypes. During the training process of compromised clients, we set the trigger and the local model to be trained alternately.

**Baselines**. To evaluate the effectiveness of BAPFL, we compare it with seven representative backdoor attack baselines: MR (Bagdasaryan et al., 2020), DBA (Xie et al., 2020), PFedBA (Lyu et al., 2024), BapFL (Ye et al., 2024), Bad-PFL (Fan et al., 2025), Chameleon (Dai & Li, 2023) and A3FL (Zhang et al., 2023a). We also assess the adaptability of BAPFL across other PFL frameworks, including FedPD (Tan et al., 2025) and FedPCL (Tan et al., 2022b).

**Defenses**. We apply various backdoor defenses, including Multi-Krum (Blanchard et al., 2017), Median (Zhang et al., 2023e), Clipping (Wang et al., 2020), Sign (Guo et al., 2023), FLAME (Nguyen et al., 2022) and Deepsight (Rieger et al., 2022).

**Metrics**. We report average main task accuracy (ACC, %) over clean samples and average attack success rate (ASR, %) over triggered samples for all benign clients' models on their test sets.

Table 1: ACC and ASR of BAPFL and baselines in PFL.

| Method | AR = 10% | | AR = 20% | | AR = 30% | | AR = 40% | |
|---|---|---|---|---|---|---|---|---|
| | ACC | ASR | ACC | ASR | ACC | ASR | ACC | ASR |
| **MNIST** | | | | | | | | |
| MR | $96.79 \pm 0.08$ | $13.29 \pm 0.12$ | $97.39 \pm 0.09$ | $24.54 \pm 0.10$ | $97.47 \pm 0.07$ | $40.27 \pm 0.11$ | $97.89 \pm 0.10$ | $50.99 \pm 0.11$ |
| DBA | $97.60 \pm 0.06$ | $38.52 \pm 0.09$ | $97.51 \pm 0.07$ | $42.44 \pm 0.08$ | $96.04 \pm 0.08$ | $49.94 \pm 0.10$ | $98.08 \pm 0.06$ | $56.40 \pm 0.09$ |
| PFedBA | $97.98 \pm 0.09$ | $30.64 \pm 0.11$ | $97.45 \pm 0.06$ | $37.67 \pm 0.14$ | $96.32 \pm 0.10$ | $48.94 \pm 0.09$ | $97.22 \pm 0.09$ | $58.12 \pm 0.13$ |
| BapFL | $97.65 \pm 0.05$ | $35.48 \pm 0.09$ | $97.40 \pm 0.07$ | $42.03 \pm 0.10$ | $97.10 \pm 0.08$ | $50.22 \pm 0.13$ | $96.80 \pm 0.05$ | $58.95 \pm 0.12$ |
| Bad-PFL | $97.30 \pm 0.07$ | $28.15 \pm 0.08$ | $97.20 \pm 0.06$ | $33.77 \pm 0.09$ | $96.95 \pm 0.07$ | $39.80 \pm 0.10$ | $96.60 \pm 0.08$ | $46.50 \pm 0.14$ |
| Chameleon | $97.85 \pm 0.05$ | $32.10 \pm 0.07$ | $97.68 \pm 0.09$ | $38.25 \pm 0.08$ | $97.55 \pm 0.05$ | $44.34 \pm 0.09$ | $97.41 \pm 0.06$ | $49.75 \pm 0.10$ |
| A3FL | $97.90 \pm 0.07$ | $34.82 \pm 0.06$ | $97.75 \pm 0.08$ | $40.90 \pm 0.07$ | $97.62 \pm 0.06$ | $47.16 \pm 0.12$ | $97.45 \pm 0.05$ | $52.60 \pm 0.13$ |
| BAPFL | $97.96 \pm 0.05$ | $\mathbf{87.14 \pm 0.10}$ | $97.85 \pm 0.06$ | $\mathbf{88.38 \pm 0.12}$ | $96.89 \pm 0.07$ | $\mathbf{88.89 \pm 0.09}$ | $96.90 \pm 0.05$ | $\mathbf{91.08 \pm 0.11}$ |
| **FEMNIST** | | | | | | | | |
| MR | $90.97 \pm 0.09$ | $13.68 \pm 0.11$ | $91.31 \pm 0.10$ | $14.18 \pm 0.09$ | $90.17 \pm 0.08$ | $18.73 \pm 0.10$ | $88.31 \pm 0.11$ | $28.52 \pm 0.12$ |
| DBA | $89.80 \pm 0.07$ | $11.71 \pm 0.10$ | $91.21 \pm 0.08$ | $17.96 \pm 0.11$ | $89.83 \pm 0.09$ | $21.10 \pm 0.12$ | $89.41 \pm 0.07$ | $41.67 \pm 0.09$ |
| PFedBA | $90.49 \pm 0.10$ | $9.43 \pm 0.11$ | $91.11 \pm 0.05$ | $17.31 \pm 0.13$ | $89.05 \pm 0.06$ | $20.63 \pm 0.11$ | $88.37 \pm 0.10$ | $40.83 \pm 0.08$ |
| BapFL | $91.20 \pm 0.06$ | $36.10 \pm 0.08$ | $90.95 \pm 0.07$ | $40.50 \pm 0.09$ | $90.30 \pm 0.07$ | $43.25 \pm 0.09$ | $89.50 \pm 0.06$ | $49.80 \pm 0.12$ |
| Bad-PFL | $90.75 \pm 0.09$ | $29.45 \pm 0.09$ | $90.60 \pm 0.06$ | $34.10 \pm 0.08$ | $90.05 \pm 0.08$ | $39.95 \pm 0.10$ | $89.00 \pm 0.07$ | $47.25 \pm 0.11$ |
| Chameleon | $91.82 \pm 0.09$ | $28.45 \pm 0.06$ | $91.50 \pm 0.08$ | $33.90 \pm 0.10$ | $91.25 \pm 0.06$ | $38.55 \pm 0.09$ | $90.85 \pm 0.08$ | $44.08 \pm 0.13$ |
| A3FL | $91.90 \pm 0.05$ | $30.25 \pm 0.07$ | $91.65 \pm 0.07$ | $36.80 \pm 0.12$ | $91.33 \pm 0.08$ | $41.90 \pm 0.10$ | $90.95 \pm 0.07$ | $47.10 \pm 0.12$ |
| BAPFL | $91.94 \pm 0.06$ | $\mathbf{87.39 \pm 0.08}$ | $91.29 \pm 0.05$ | $\mathbf{88.48 \pm 0.09}$ | $90.55 \pm 0.07$ | $\mathbf{89.19 \pm 0.11}$ | $89.18 \pm 0.06$ | $\mathbf{89.23 \pm 0.10}$ |
| **CIFAR-10** | | | | | | | | |
| MR | $66.10 \pm 0.11$ | $11.32 \pm 0.10$ | $63.93 \pm 0.10$ | $13.08 \pm 0.12$ | $60.75 \pm 0.09$ | $13.36 \pm 0.11$ | $66.31 \pm 0.10$ | $13.81 \pm 0.09$ |
| DBA | $65.97 \pm 0.08$ | $10.25 \pm 0.09$ | $60.31 \pm 0.09$ | $10.63 \pm 0.14$ | $65.71 \pm 0.08$ | $13.48 \pm 0.15$ | $66.44 \pm 0.09$ | $13.59 \pm 0.10$ |
| PFedBA | $65.78 \pm 0.09$ | $7.11 \pm 0.08$ | $63.82 \pm 0.06$ | $8.91 \pm 0.14$ | $61.78 \pm 0.12$ | $13.59 \pm 0.08$ | $51.41 \pm 0.05$ | $19.27 \pm 0.13$ |
| BapFL | $64.80 \pm 0.06$ | $12.10 \pm 0.09$ | $63.95 \pm 0.07$ | $20.40 \pm 0.10$ | $62.70 \pm 0.06$ | $26.30 \pm 0.11$ | $61.50 \pm 0.07$ | $29.95 \pm 0.12$ |
| Bad-PFL | $64.10 \pm 0.09$ | $9.75 \pm 0.13$ | $63.20 \pm 0.08$ | $18.50 \pm 0.09$ | $62.10 \pm 0.07$ | $23.60 \pm 0.10$ | $60.95 \pm 0.06$ | $26.50 \pm 0.08$ |
| Chameleon | $62.46 \pm 0.08$ | $12.86 \pm 0.09$ | $61.90 \pm 0.08$ | $16.55 \pm 0.12$ | $61.75 \pm 0.09$ | $21.47 \pm 0.14$ | $61.50 \pm 0.08$ | $25.90 \pm 0.13$ |
| A3FL | $62.35 \pm 0.06$ | $14.64 \pm 0.10$ | $61.80 \pm 0.09$ | $18.90 \pm 0.10$ | $61.60 \pm 0.08$ | $23.56 \pm 0.12$ | $61.35 \pm 0.09$ | $27.85 \pm 0.11$ |
| BAPFL | $62.38 \pm 0.07$ | $\mathbf{77.38 \pm 0.12}$ | $61.47 \pm 0.08$ | $\mathbf{77.78 \pm 0.11}$ | $60.93 \pm 0.06$ | $\mathbf{78.20 \pm 0.10}$ | $60.83 \pm 0.07$ | $\mathbf{82.00 \pm 0.12}$ |
| **CIFAR-100** | | | | | | | | |
| MR | $67.31 \pm 0.16$ | $5.12 \pm 0.22$ | $66.41 \pm 0.15$ | $8.45 \pm 0.18$ | $68.96 \pm 0.20$ | $9.63 \pm 0.25$ | $67.59 \pm 0.17$ | $10.58 \pm 0.21$ |
| DBA | $67.10 \pm 0.18$ | $8.56 \pm 0.20$ | $66.80 \pm 0.17$ | $10.42 \pm 0.21$ | $66.65 \pm 0.19$ | $12.15 \pm 0.23$ | $67.45 \pm 0.18$ | $13.82 \pm 0.22$ |
| PFedBA | $67.22 \pm 0.19$ | $7.30 \pm 0.22$ | $67.92 \pm 0.14$ | $11.65 \pm 0.20$ | $66.96 \pm 0.18$ | $13.70 \pm 0.24$ | $67.67 \pm 0.16$ | $15.48 \pm 0.20$ |
| BapFL | $67.10 \pm 0.18$ | $12.35 \pm 0.24$ | $67.45 \pm 0.16$ | $19.35 \pm 0.22$ | $67.38 \pm 0.17$ | $22.12 \pm 0.20$ | $67.21 \pm 0.18$ | $24.25 \pm 0.23$ |
| Bad-PFL | $67.20 \pm 0.17$ | $11.25 \pm 0.19$ | $66.99 \pm 0.15$ | $15.10 \pm 0.21$ | $67.50 \pm 0.16$ | $19.56 \pm 0.23$ | $67.80 \pm 0.17$ | $20.12 \pm 0.19$ |
| Chameleon | $67.28 \pm 0.18$ | $10.35 \pm 0.18$ | $67.05 \pm 0.17$ | $13.85 \pm 0.22$ | $66.95 \pm 0.18$ | $17.25 \pm 0.19$ | $66.85 \pm 0.15$ | $20.10 \pm 0.20$ |
| A3FL | $67.32 \pm 0.19$ | $11.80 \pm 0.20$ | $67.15 \pm 0.16$ | $15.25 \pm 0.19$ | $67.00 \pm 0.17$ | $18.95 \pm 0.20$ | $66.90 \pm 0.16$ | $21.75 \pm 0.19$ |
| BAPFL | $67.38 \pm 0.15$ | $\mathbf{75.67 \pm 0.18}$ | $68.67 \pm 0.16$ | $\mathbf{76.22 \pm 0.15}$ | $67.17 \pm 0.19$ | $\mathbf{77.81 \pm 0.17}$ | $68.02 \pm 0.14$ | $\mathbf{79.82 \pm 0.16}$ |

## 5.2 MAIN RESULTS

### 5.2.1 COMPARISONS BETWEEN BAPFL AND BASELINES

In the PFL framework FedProto, we compare the performance of BAPFL with baselines, i.e., MR, DBA, PFedBA, BapFL, Bad-PFL, Chameleon, and A3FL, under varying attack rates on MNIST, FEMNIST, CIFAR-10, and CIFAR-100. As shown in Table 1, BAPFL consistently achieves the highest ASR across all settings, while maintaining comparable or even higher ACC. Notably, BAPFL improves ASR by 33%–75% over baselines, demonstrating its superior effectiveness and stealthiness in PFL. This is attributed to BAPFL's unique prototype poisoning strategy tailored for PFL, which misleads the optimization of the global prototype, and its multi-trigger optimization mechanism that adapts to the heterogeneous label distribution across clients.

### 5.2.2 THE ATTACK PERFORMANCE OF BAPFL AGAINST ADVANCED DEFENSES

To further assess the stealthiness of BAPFL, we evaluate its performance under robust PFL with several advanced defense strategies, namely Multi-Krum (Blanchard et al., 2017), Median (Zhang et al., 2023e), Clipping (Wang et al., 2020), Sign (Guo et al., 2023), FLAME (Nguyen et al., 2022), and Deepsight (Rieger et al., 2022). Table 2 presents the results of BAPFL on MNIST, FEMNIST, CIFAR-10, and CIFAR-100 under these advanced defenses. We observe that these defense strategies can mitigate the attack effects of BAPFL. Among them, FLAME achieves the strongest defensive effects. However, even with FLAME, BAPFL still achieves at least 60% ASR while maintaining a high ACC. This indicates that existing robust aggregation strategies have only limited effectiveness on BAPFL. The key reason is that BAPFL leverages PPS to precisely control both the magnitude and direction of poisoned prototypes, rendering them indistinguishable from benign ones (See Appendix F for further analysis).

Given these limitations, we believe that specialized defense mechanisms are needed to effectively counter BAPFL. For instance, the server can maintain the historical information of class prototypes uploaded by each client, and visualize the optimization paths rather than the current round of these

Table 2: ACC and ASR of BAPFL under advanced defenses across different attack rates.

| Defense | AR=10% | | AR=20% | | AR=30% | | AR=40% | |
|---|---|---|---|---|---|---|---|---|
| | ACC | ASR | ACC | ASR | ACC | ASR | ACC | ASR |
| MNIST | | | | | | | | |
| Multi-Krum | 97.06 ± 0.08 | 66.81 ± 0.11 | 97.20 ± 0.06 | 67.30 ± 0.12 | 97.34 ± 0.09 | 71.02 ± 0.13 | 97.16 ± 0.07 | 74.64 ± 0.14 |
| Median | 97.36 ± 0.10 | 70.89 ± 0.13 | 98.20 ± 0.12 | 73.39 ± 0.14 | 97.56 ± 0.09 | 75.26 ± 0.12 | 97.90 ± 0.08 | 78.93 ± 0.15 |
| Clipping | 97.89 ± 0.11 | 68.09 ± 0.10 | 97.52 ± 0.07 | 73.74 ± 0.12 | 97.24 ± 0.12 | 76.92 ± 0.13 | 98.39 ± 0.09 | 77.08 ± 0.11 |
| Sign | 97.57 ± 0.06 | 77.85 ± 0.13 | 97.42 ± 0.08 | 78.10 ± 0.12 | 97.36 ± 0.05 | 79.46 ± 0.14 | 97.76 ± 0.09 | 80.53 ± 0.15 |
| FLAME | 98.01 ± 0.12 | 63.58 ± 0.20 | 98.06 ± 0.14 | 65.48 ± 0.18 | 97.82 ± 0.13 | 69.84 ± 0.19 | 98.28 ± 0.12 | 70.21 ± 0.21 |
| Deepsight | 97.91 ± 0.15 | 64.63 ± 0.22 | 97.87 ± 0.13 | 66.23 ± 0.19 | 98.19 ± 0.12 | 71.21 ± 0.18 | 98.37 ± 0.11 | 72.99 ± 0.17 |
| FEMNIST | | | | | | | | |
| Multi-Krum | 90.86 ± 0.09 | 63.42 ± 0.13 | 90.69 ± 0.11 | 64.33 ± 0.12 | 91.27 ± 0.10 | 71.27 ± 0.15 | 90.71 ± 0.08 | 74.03 ± 0.14 |
| Median | 91.85 ± 0.07 | 63.32 ± 0.10 | 92.75 ± 0.09 | 66.86 ± 0.14 | 92.58 ± 0.06 | 70.67 ± 0.13 | 92.90 ± 0.10 | 75.72 ± 0.12 |
| Clipping | 91.73 ± 0.12 | 67.57 ± 0.09 | 92.32 ± 0.08 | 69.33 ± 0.15 | 91.66 ± 0.11 | 72.30 ± 0.14 | 92.14 ± 0.07 | 74.77 ± 0.13 |
| Sign | 91.98 ± 0.05 | 76.90 ± 0.12 | 90.44 ± 0.09 | 77.78 ± 0.14 | 90.75 ± 0.07 | 78.45 ± 0.15 | 89.98 ± 0.11 | 80.06 ± 0.13 |
| FLAME | 90.12 ± 0.13 | 61.60 ± 0.15 | 89.89 ± 0.14 | 63.51 ± 0.17 | 89.67 ± 0.15 | 67.13 ± 0.19 | 89.92 ± 0.12 | 71.41 ± 0.22 |
| Deepsight | 90.87 ± 0.12 | 62.71 ± 0.18 | 90.77 ± 0.10 | 65.40 ± 0.16 | 90.82 ± 0.13 | 70.27 ± 0.17 | 91.15 ± 0.14 | 72.40 ± 0.20 |
| CIFAR-10 | | | | | | | | |
| Multi-Krum | 57.15 ± 0.07 | 74.69 ± 0.14 | 57.00 ± 0.11 | 76.37 ± 0.12 | 55.43 ± 0.09 | 78.07 ± 0.13 | 55.16 ± 0.08 | 78.23 ± 0.15 |
| Median | 58.30 ± 0.12 | 73.32 ± 0.11 | 57.44 ± 0.10 | 75.97 ± 0.13 | 57.85 ± 0.09 | 76.12 ± 0.08 | 58.84 ± 0.11 | 78.26 ± 0.15 |
| Clipping | 57.76 ± 0.06 | 75.94 ± 0.13 | 58.49 ± 0.08 | 76.33 ± 0.14 | 57.90 ± 0.12 | 77.41 ± 0.09 | 57.70 ± 0.07 | 79.56 ± 0.13 |
| Sign | 58.18 ± 0.11 | 75.71 ± 0.12 | 59.49 ± 0.09 | 76.64 ± 0.14 | 58.75 ± 0.07 | 79.30 ± 0.15 | 57.56 ± 0.10 | 80.57 ± 0.13 |
| FLAME | 57.55 ± 0.15 | 71.36 ± 0.21 | 58.60 ± 0.17 | 73.42 ± 0.18 | 58.11 ± 0.16 | 75.95 ± 0.22 | 57.41 ± 0.14 | 77.21 ± 0.19 |
| Deepsight | 59.54 ± 0.14 | 72.60 ± 0.20 | 58.98 ± 0.15 | 74.63 ± 0.22 | 59.50 ± 0.13 | 76.21 ± 0.18 | 59.39 ± 0.12 | 77.58 ± 0.20 |
| CIFAR-100 | | | | | | | | |
| Multi-Krum | 61.19 ± 0.20 | 70.42 ± 0.25 | 61.05 ± 0.18 | 72.38 ± 0.26 | 60.75 ± 0.21 | 74.91 ± 0.24 | 60.45 ± 0.19 | 75.12 ± 0.27 |
| Median | 62.21 ± 0.16 | 68.95 ± 0.21 | 62.05 ± 0.15 | 70.82 ± 0.23 | 61.85 ± 0.17 | 72.90 ± 0.22 | 61.70 ± 0.16 | 73.92 ± 0.23 |
| Clipping | 61.75 ± 0.18 | 69.30 ± 0.22 | 61.60 ± 0.17 | 71.15 ± 0.24 | 61.40 ± 0.19 | 73.58 ± 0.23 | 61.35 ± 0.18 | 74.85 ± 0.25 |
| Sign | 62.83 ± 0.15 | 72.25 ± 0.24 | 62.55 ± 0.14 | 74.58 ± 0.25 | 62.35 ± 0.16 | 76.62 ± 0.26 | 62.10 ± 0.15 | 77.54 ± 0.27 |
| FLAME | 61.95 ± 0.19 | 65.91 ± 0.23 | 62.00 ± 0.18 | 66.85 ± 0.24 | 61.85 ± 0.20 | 69.92 ± 0.26 | 61.65 ± 0.19 | 70.25 ± 0.27 |
| Deepsight | 62.35 ± 0.17 | 67.85 ± 0.21 | 62.20 ± 0.16 | 69.91 ± 0.23 | 62.05 ± 0.18 | 71.84 ± 0.24 | 61.90 ± 0.17 | 72.98 ± 0.25 |

Table 3: BAPFL performance in various PFL frameworks.

| Method | AR = 10% | | AR = 20% | | AR = 30% | | AR = 40% | |
|---|---|---|---|---|---|---|---|---|
| | ACC | ASR | ACC | ASR | ACC | ASR | ACC | ASR |
| FedPCL | 49.11 ± 0.10 | 72.91 ± 0.10 | 48.64 ± 0.11 | 75.89 ± 0.12 | 48.61 ± 0.10 | 78.78 ± 0.11 | 49.81 ± 0.12 | 81.82 ± 0.12 |
| FedPD | 97.87 ± 0.08 | 65.11 ± 0.09 | 97.91 ± 0.07 | 70.56 ± 0.10 | 96.97 ± 0.09 | 77.74 ± 0.11 | 97.28 ± 0.08 | 79.40 ± 0.10 |

prototypes to identify poisoned prototypes. This is because the single-round update of poisoned prototypes is minimal and difficult to distinguish from the update of benign prototypes, while the optimization paths of benign and poisoned prototypes are clearly different. Specifically, in PFL, the poisoned prototypes share a consistent optimization objective, which aims to pull the global prototype away from the trigger prototype, rather than move towards the global prototype. In contrast, the benign prototypes aims to continually move towards the global prototype. Additionally, benign clients can further mitigate the effects of BAPFL by fine-tuning their local models on clean datasets and correcting the misled benign prototypes.

### 5.2.3 INTEGRATE BAPFL INTO DIFFERENT PFL FRAMEWORKS

To demonstrate the adaptability of BAPFL, we evaluate its effectiveness against two other representative PFL frameworks: FedPCL and FedPD. FedPCL employs a contrastive loss to enhance prototype alignment. FedPD adopts robust aggregation based on cosine similarity and encourages inter-class prototype separation. We apply BAPFL to FedPCL on the OFFICE-10 dataset (Gong et al., 2012), and to FedPD on MNIST, respectively. The results are shown in Table 3. We observe that as the attack rate increases, the ASR of BAPFL in FedPCL increases from 72.91% to 81.82%, while ACC remains stable. This confirms the vulnerability of FedPCL to our attack. Moreover, although FedPD adopts robust aggregation, BAPFL still achieves 65.11%-79.4% ASR, demonstrating its ability to bypass FedPD's defense.

### 5.2.4 DATA HETEROGENEITY

To evaluate the robustness of BAPFL under varying degrees of data heterogeneity, we simulate different data heterogeneity scenarios by adjusting the values of $p$, $q$, and $\beta$ across clients. Specifically, for MNIST, we fix $q = 100$, $\beta = 0.5$, and vary $p$ from 3 to 7. For FEMNIST, we fix $p = 5$, $\beta = 0.5$, and vary $q$ from 40 to 120. For CIFAR-10, we fix $p = 5$, $q = 100$, and vary $\beta$ over $\{0.1, 0.3, 0.5, 0.7, 0.9\}$. Then, we evaluate the ASR of our BAPFL method under these settings. The

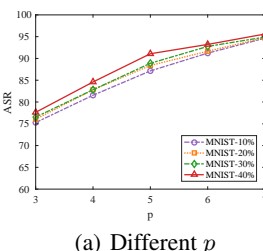 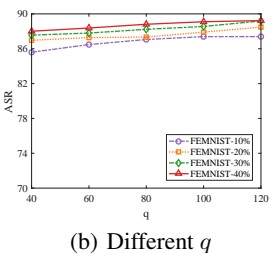 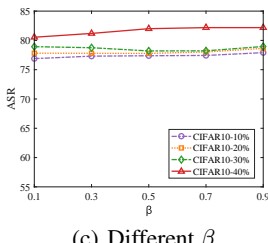

(a) Different $p$        (b) Different $q$        (c) Different $\beta$

Figure 5: The ASR of BAPFL under varying degrees of data heterogeneity.

Table 4: Ablation results of BAPFL on benchmark datasets under different attack rates.

| Method | AR = 10% | | AR = 20% | | AR = 30% | | AR = 40% | |
|---|---|---|---|---|---|---|---|---|
| | ACC | ASR | ACC | ASR | ACC | ASR | ACC | ASR |
| **MNIST** | | | | | | | | |
| DBA | $97.60 \pm 0.11$ | $38.52 \pm 0.12$ | $97.51 \pm 0.09$ | $42.44 \pm 0.10$ | $96.04 \pm 0.10$ | $49.94 \pm 0.11$ | $98.08 \pm 0.08$ | $56.40 \pm 0.13$ |
| DBA+PPS | $98.05 \pm 0.10$ | $62.15 \pm 0.11$ | $97.98 \pm 0.12$ | $69.43 \pm 0.09$ | $97.72 \pm 0.08$ | $71.50 \pm 0.12$ | $97.48 \pm 0.11$ | $75.64 \pm 0.10$ |
| PPS+TOM | $97.96 \pm 0.05$ | $\mathbf{87.14 \pm 0.10}$ | $97.85 \pm 0.06$ | $\mathbf{88.38 \pm 0.12}$ | $96.89 \pm 0.07$ | $\mathbf{88.89 \pm 0.09}$ | $96.90 \pm 0.05$ | $\mathbf{91.08 \pm 0.11}$ |
| **FEMNIST** | | | | | | | | |
| DBA | $89.80 \pm 0.09$ | $11.71 \pm 0.10$ | $91.21 \pm 0.12$ | $17.96 \pm 0.11$ | $89.83 \pm 0.10$ | $21.10 \pm 0.12$ | $89.41 \pm 0.08$ | $41.67 \pm 0.13$ |
| DBA+PPS | $89.80 \pm 0.11$ | $60.25 \pm 0.09$ | $90.09 \pm 0.08$ | $69.72 \pm 0.12$ | $89.72 \pm 0.09$ | $72.72 \pm 0.10$ | $88.91 \pm 0.10$ | $73.58 \pm 0.11$ |
| PPS+TOM | $91.94 \pm 0.06$ | $\mathbf{87.39 \pm 0.08}$ | $91.29 \pm 0.05$ | $\mathbf{88.48 \pm 0.09}$ | $90.55 \pm 0.07$ | $\mathbf{89.19 \pm 0.11}$ | $89.18 \pm 0.06$ | $\mathbf{89.23 \pm 0.10}$ |
| **CIFAR-10** | | | | | | | | |
| DBA | $65.97 \pm 0.08$ | $10.25 \pm 0.09$ | $60.31 \pm 0.10$ | $10.63 \pm 0.12$ | $65.71 \pm 0.09$ | $13.48 \pm 0.11$ | $66.44 \pm 0.11$ | $13.59 \pm 0.10$ |
| DBA+PPS | $65.59 \pm 0.09$ | $45.64 \pm 0.11$ | $61.86 \pm 0.08$ | $47.85 \pm 0.10$ | $61.87 \pm 0.11$ | $49.93 \pm 0.09$ | $65.16 \pm 0.10$ | $50.69 \pm 0.11$ |
| PPS+TOM | $62.38 \pm 0.07$ | $\mathbf{77.38 \pm 0.12}$ | $61.47 \pm 0.08$ | $\mathbf{77.78 \pm 0.11}$ | $60.93 \pm 0.06$ | $\mathbf{78.20 \pm 0.10}$ | $60.83 \pm 0.07$ | $\mathbf{82.00 \pm 0.12}$ |
| **CIFAR-100** | | | | | | | | |
| DBA | $67.10 \pm 0.18$ | $8.56 \pm 0.20$ | $66.80 \pm 0.17$ | $10.42 \pm 0.21$ | $66.65 \pm 0.19$ | $12.15 \pm 0.23$ | $67.45 \pm 0.18$ | $13.82 \pm 0.22$ |
| DBA+PPS | $67.52 \pm 0.16$ | $37.68 \pm 0.16$ | $68.21 \pm 0.14$ | $39.83 \pm 0.17$ | $67.73 \pm 0.17$ | $42.14 \pm 0.19$ | $67.88 \pm 0.15$ | $44.30 \pm 0.20$ |
| PPS+TOM | $67.38 \pm 0.15$ | $\mathbf{75.67 \pm 0.18}$ | $68.67 \pm 0.16$ | $\mathbf{76.22 \pm 0.15}$ | $67.17 \pm 0.19$ | $\mathbf{77.81 \pm 0.17}$ | $68.02 \pm 0.14$ | $\mathbf{79.82 \pm 0.16}$ |

experimental results in Figure 5 show that BAPFL consistently achieves ASR of at least 75% across all heterogeneous settings, demonstrating its strong robustness.

## 5.3 ABLATION STUDY

To evaluate the effectiveness of each component in BAPFL, we conduct an ablation study of BAPFL on three datasets, i.e., MNIST, FEMNIST, CIFAR-10, and CIFAR-100. Specifically, we examine the individual contributions of the PPS and TOM in BAPFL. The experimental results under different datasets and various attack rates are summarized in Table 4. Across all datasets, BAPFL(PPS+TOM) consistently achieves the highest ASR with minimal impact on ACC. Removing either component significantly reduces ASR of BAPFL. For example, in the ablation study based on MNIST, when AR is 20%, BAPFL(PPS+TOM) achieves 88.38% ASR, while BAPFL(DBA+PPS) drops ASR to 69.43%, and BAPFL(DBA) alone achieves only 42.44%. Similar trends are observed in results based on FEMNIST, CIFAR-10 and CIFAR-100. These results highlight that both PPS and TOM are essential for enhancing BAPFL's effectiveness. We also provide a case study in Appendix J to visualize the effects of PPS and TOM. Additionally, the computation and communication overhead introduced by BAPFL(PPS+TOM) is analyzed in Appendix K, showing the low overhead of BAPFL.

## 6 CONCLUSIONS AND FUTURE WORK

In this paper, we investigate the problems of applying existing backdoor attacks in PFL and propose a novel and effective backdoor attack method BAPFL. By carefully designing poisoned prototypes and optimizing specific triggers for target labels, BAPFL successfully induces targeted misclassifications in benign models while evading detection. Comprehensive evaluations across diverse datasets and PFL frameworks demonstrate that BAPFL significantly improves ASR with negligible performance degradation on main tasks. BAPFL underscores the need for stronger defenses in PFL and provides insights into designing secure and trustworthy PFL systems. In future work, we intend to extend our methodology to other FL frameworks that adopt non-gradient-based aggregation strategies.

## 7 ETHICS STATEMENT

This paper presents an attack method that undermines the trustworthiness of federated learning. Although this attack method may seem harmful, we strongly believe that the benefits of publishing this paper outweigh the drawbacks. Specifically, this attack method can motivate researchers to explore more effective defense strategies, serve as an assessment tool for testing the trustworthiness of federated learning, and raise awareness of potential threats faced by users implementing federated learning in real-world scenarios.

## 8 REPRODUCIBILITY STATEMENT

The source codes are available in https://anonymous.4open.science/r/BAPFL-C420/.

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

## A  USE OF LARGE LANGUAGE MODELS (LLMs)

In this paper, we utilize LLMs as an auxiliary tool to assist in checking grammar and spelling errors.

## B  EXPERIMENTAL SETTINGS

Our experiments are conducted on a small-scale local network consisting of machines equipped with Intel Xeon CPU E5-1650 v4 @ 3.60GHz, 64GB RAM, and NVIDIA GTX 4090 GPUs. All experiments are implemented in Python. The backdoor attack is implemented based on the code of PFedBA (Lyu et al., 2024), and the attack rate varies from $\{10\%, 20\%, 30\%, 40\%\}$. The compromised clients train triggers for the target labels over 50 local rounds and embed them into their local data to train local models. During the training process of compromised clients, we set the trigger and the local model to be trained alternately. For all experiments, the model information is detailed in Table 5, and the training settings are described in Section 5.1.

Table 5: Dataset and model architecture

| Dataset | Labels | Image size | Training/Test images | Model |
|---------|--------|-----------|---------------------|-------|
| MNIST | 10 | 1*28*28 | 60k/10k | 2Conv + 2Fc |
| FEMNIST | 10 | 1*128*128 | 22k/3k | 2Conv + 2Fc |
| CIFAR-10 | 10 | 3*32*32 | 50k/10k | ResNet18 (He et al., 2016) |
| CIFAR-100 | 100 | 3*32*32 | 50k/10k | ResNet101 (He et al., 2016) |

## C  THE PSEUDO-CODE OF APPLYING BAPFL TO PFL

---

**Algorithm 1** PFL process with BAPFL Attack

---

1: **Server Executes:**
2: Initialize global prototypes $\bar{P} = \{\bar{P}^{(k)}\}_{k=1,2,...}$
3: **while** *the current training round $r \leq$ the final round* **do**
4:     Broadcast $\bar{P}$ to clients for local training
5:     Aggregate the local prototypes of clients to update $\bar{P}$
6: **end while**
7: **Client Executes:**
8: **if** *this client is compromised* **then**
9:     /*Execute TOM*/
10:    Triggers ← Download the trigger network from the adversary and train it with $\bar{P}$ based on Equation equation 8
11:    /*Execute PPS*/
12:    Select the top-$K$ samples with the highest attack value based on Equation equation 6
13:    Train $f_\theta$ according to Equation equation 5
14:    Poisoned prototypes $P_c$ ← Flip trigger prototypes
15: **else**
16:    Train $P_c$ and $f_\theta$ with $\bar{P}$ according to Equation equation 4
17: **end if**
18: **return** the new prototypes $P_c$ to the server

---

## D  VISUAL EXAMPLES OF TRIGGER-EMBEDDED SAMPLES

In this appendix, we present visual examples of trigger-embedded samples generated by our proposed BAPFL method across the three datasets used in our experiments. Specifically, we selected two images with embedded triggers from each dataset to illustrate the stealthiness of the backdoor

attack. As shown in Figures 6, the triggers generated by BAPFL introduce only minimal and visually imperceptible modifications, demonstrating the high stealthiness of our attack.

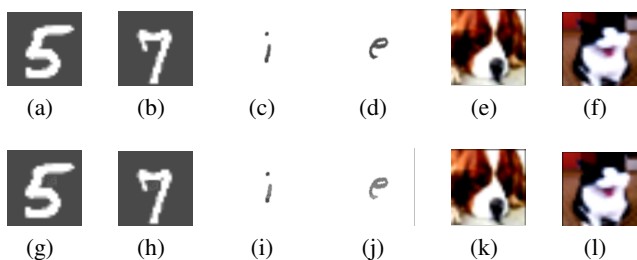

Figure 6: Original images (a-f) vs. trigger-embedded images (g-l).

# E  THEORETICAL ANALYSIS OF BAPFL

In this appendix, we formally analyze how the proposed BAPFL enhances backdoor effectiveness in prototype-based federated learning (PFL).

## E.1  EFFECTIVENESS OF PPS

We first introduce a key assumption that underlies prototype-based classification in PFL.

**Assumption 1.** In a well-trained PFL model, a sample $x$ is classified as label $k$ if its extracted feature $\phi(x)$ is closer (in $\ell_2$ distance) to the global prototype $\bar{P}^{(k)}$ than to any other prototype. That is,

$$\text{label}(x) = \arg\min_k \|\phi(x) - \bar{P}^{(k)}\|_2. \tag{1}$$

Under this assumption, the attack success rate (ASR) is increased if the features of trigger-embedded samples $\phi(T_{y_t}(x))$ are misaligned with the global prototype $\bar{P}^{(y)}$ of the original label and become closer to other class prototypes. We now show that PPS increases this misalignment by manipulating the process of prototype aggregation.

**Theorem 1.** The PPS increases the misclassification probability of trigger-embedded samples in benign models by poisoning the global prototype aggregation. Specifically, it manipulates the global prototype $\bar{P}^{(k)}$ of class $k$ to deviate from the trigger prototype $P_{tr}^{(k)}$, thereby misleading the optimization of benign prototypes and increasing the distance between the benign prototype $P_c^{(k)}$ and $P_{tr}^{(k)}$.

*Proof:* $\bar{P}^{(k)}$ is computed as the average of clients' local prototypes, i.e.,

$$\bar{P}^{(k)} = \frac{1}{C}\sum_{c=1}^{C} P_c^{(k)}. \tag{2}$$

The malicious client $c^*$ uploads a poisoned prototype $P_{c^*}^{(k)}$ defined as:

$$P_{c^*}^{(k)} = 2 \cdot P_{proj} - P_{tr}^{(k)}, \tag{3}$$

where

$$P_{proj} = \frac{\bar{P}^{(k)} \cdot P_{tr}^{(k)}}{\bar{P}^{(k)} \cdot \bar{P}^{(k)}} \cdot \bar{P}^{(k)}. \tag{4}$$

This poisoned prototype $P_{c^*}^{(k)}$ is the reflection of $P_{tr}^{(k)}$ with respect to the projection point $P_{proj}$ on $\bar{P}^{(k)}$, thus intentionally pushing the aggregated prototype away from the direction of $P_{tr}^{(k)}$. Let

$C_b$ denote the set of benign clients and $C_m$ the set of malicious clients. The new aggregated global prototype becomes:

$$\bar{P}_{\text{new}}^{(k)} = \frac{1}{|C_b| + |C_m|} \left( \sum_{c \in C_b} P_c^{(k)} + \sum_{c^* \in C_m} P_{c^*}^{(k)} \right). \tag{5}$$

Since the poisoned prototypes are reflected points away from $P_{tr}^{(k)}$, the vector $\bar{P}_{\text{new}}^{(k)} - P_{tr}^{(k)}$ increases in magnitude compared to $\bar{P}^{(k)} - P_{tr}^{(k)}$, i.e.,

$$\|\bar{P}_{\text{new}}^{(k)} - P_{tr}^{(k)}\|_2 > \|\bar{P}^{(k)} - P_{tr}^{(k)}\|_2. \tag{6}$$

In the subsequent training rounds, benign clients optimize their local prototypes $P_c^{(k)}$ to minimize the consistency loss $\mathcal{L}_P$ with the (now biased) global prototype:

$$\mathcal{L}_P = \|P_c^{(k)} - \bar{P}_{\text{new}}^{(k)}\|_2. \tag{7}$$

Thus, $P_c^{(k)}$ is continuously pulled toward $\bar{P}_{\text{new}}^{(k)}$, and consequently, $\|P_c^{(k)} - P_{tr}^{(k)}\|_2$ increases over training rounds. According to *Assumption 1*, for a clean sample $x$, if its feature $\phi(x)$ approximates $P_c^{(k)}$, then the classification result of this sample $x$ is:

$$k = \arg\min_j \|\phi(x) - \bar{P}_{\text{new}}^{(j)}\|_2. \tag{8}$$

For the corresponding trigger-embedded sample $x^*$ with target label $y_t$, its feature $\phi(x^*)$ approximates $P_{tr}^{(k)}$. Since $\|P_c^{(k)} - P_{tr}^{(k)}\|$ becomes larger due to PPS, the probability that $\phi(x^*)$ is closest to $\bar{P}_{\text{new}}^{(k)}$ decreases, i.e.,

$$\Pr\left[\arg\min_j \|\phi(x^*) - \bar{P}_{\text{new}}^{(j)}\|_2 = k\right] \downarrow. \tag{9}$$

Therefore, PPS increases the misclassification probability of trigger-embedded samples in benign models.

### E.2 EFFECTIVENESS OF TOM

We now analyze how TOM increases the ASR in PFL. Specifically, by aligning trigger prototypes with the global prototypes of target labels that overlap with local label spaces, TOM increases the probability that the trigger-target label mapping is unintentionally activated in benign clients.

**Assumption 2.** For a trigger-embedded sample $x^*$ with target label $y_t$, its classification is determined by the proximity of its feature $\phi(T_{y_t}(x))$ to the global prototype $\bar{P}^{(k)}$:

$$\text{label}(T_{y_t}(x)) = \arg\min_k \|\phi(T_{y_t}(x)) - \bar{P}^{(k)}\|_2. \tag{10}$$

**Theorem 2.** TOM increases the probability that the trigger-target label mapping is unintentionally activated by benign models.

*Proof:* Let $Y_t$ denote the set of target labels chosen by the attacker, where:

$$Y_t = \bigcup_{c \in C_b} \mathcal{Y}_c, \tag{11}$$

and $\mathcal{Y}_c$ is the local label space of benign client $c$. $Y_t$ maximizes the probability that any benign client $c$ has $y_t \in \mathcal{Y}_c$ for some $y_t \in Y_t$. For each target label $y_t \in Y_t$, TOM optimizes a dedicated trigger pattern $(\delta_{y_t}, M_{y_t})$ to construct a trigger function:

$$T_{y_t}(x) = (1 - M_{y_t}) \odot x + M_{y_t} \odot \delta_{y_t}. \tag{12}$$

TOM jointly optimizes $(\delta_{y_t}, M_{y_t})$ to minimize the loss $\mathcal{L}_{\text{trigger}}$. After training, $P_{tr}^{(y_t)} \approx \bar{P}^{(y_t)}$. Now consider a benign client $c$ such that $y_t \in \mathcal{Y}_c$. Suppose the trigger-embedded sample $T_{y_t}(x)$ is injected into $c$'s testing batch, TOM increases the likelihood that:

$$\phi(T_{y_t}(x)) \approx P_{tr}^{(y_t)} \approx \bar{P}^{(y_t)}. \tag{13}$$

Therefore, under *Assumption 2*, the probability that a benign model classifies $T_{y_t}(x)$ as $y_t$ increases:

$$\Pr\left[\arg\min_k \|\phi(T_{y_t}(x)) - \bar{P}^{(k)}\|_2 = y_t\right] \uparrow. \tag{14}$$

Hence, TOM increases the chance of "unintentional backdoor activation" across benign clients.

In conclusion, our theoretical analysis demonstrates that the integration of PPS and TOM enables BAPFL to effectively enhance the the ASR in PFL.

# F  THEORETICAL PROOF: BAPFL BYPASSES OUTLIER-BASED ROBUST AGGREGATION STRATEGIES

Let $F : \{P_c^{(k)}\}_{c=1}^C \to \bar{P}^{(k)}$ be a robust aggregation function that computes the global prototype $\bar{P}^{(k)}$ from local prototypes $P_c^{(k)}$ uploaded by clients. Outlier-based defenses (e.g., Multi-Krum) rely on *similarity-based or distance-based filtering*, rejecting prototypes that deviate significantly from the majority distribution. Let $P_{c^*}^{(k)}$ denote the malicious prototype generated by BAPFL's prototype poisoning strategy (PPS).

**Theorem 1.** BAPFL can successfully evade any similarity-based or distance-based outlier detection strategy $F$ by ensuring that the malicious prototype $P_{c^*}^{(k)}$ generated by PPS stays within an arbitrarily small neighborhood of the benign prototype distribution.

**Proof.** Assume benign local prototypes follow

$$P_c^{(k)} = \mu^{(k)} + \epsilon_c, \qquad \|\epsilon_c\|_2 \leq \sigma. \tag{15}$$

The outlier-based defenses (e.g., Multi-Krum) reject any prototype with deviation exceeding

$$\|P_c^{(k)} - \mu^{(k)}\|_2 > \tau, \qquad \tau = O(\sigma). \tag{16}$$

In BAPFL, the trigger used is optimized iteratively and designed to be stealthy, meaning that the embedded trigger introduces only minimal perturbation to the original image. Due to this imperceptibility, the trigger barely affects the extracted feature representation. Consequently, the trigger prototype $P_{tr}^{(k)}$ stays close to both the benign prototype and the global prototype $\bar{P}^{(k)}$, i.e.,

$$\|P_{tr}^{(k)} - \bar{P}^{(k)}\|_2 \approx \|P_c^{(k)} - \bar{P}^{(k)}\|_2 \leq \tau - \delta, \qquad \delta \ll 1. \tag{17}$$

The poisoned prototype $P_{c^*}^{(k)}$ is constructed as the axis-symmetric version of $P_{tr}^{(k)}$ with respect to the global benign prototype, i.e.,

$$P_{c^*}^{(k)} = 2 \cdot P_{\text{proj}} - P_{tr}^{(k)}, \tag{18}$$

where $P_{\text{proj}}$ is the projection of $P_{tr}^{(k)}$ onto $\bar{P}^{(k)}$:

$$P_{\text{proj}} = \frac{\bar{P}^{(k)} \cdot P_{tr}^{(k)}}{\|\bar{P}^{(k)}\|_2^2} \cdot \bar{P}^{(k)}. \tag{19}$$

This geometric construction ensures:

$$\|P_{c^*}^{(k)} - \bar{P}^{(k)}\|_2 = \|P_{tr}^{(k)} - \bar{P}^{(k)}\|_2. \tag{20}$$

Since outlier-based defenses cannot detect $P_{tr}^{(k)}$, the axis-symmetric poisoned prototype $P_{c^*}^{(k)}$, which preserves the same distance to $\bar{P}^{(k)}$, is also undetectable. That is,

$$\|P_{c^*}^{(k)} - \bar{P}^{(k)}\|_2 \leq \tau - \delta, \tag{21}$$

Thus, the poisoned prototype remains within the inlier region of the robust aggregation function $F$ and cannot be filtered out by any similarity-based or distance-based defense strategy. Hence, $F$ cannot detect or remove $P_{c^*}^{(k)}$.

Table 6: Comparison of different sample selection strategies in terms of benign ACC and ASR.

| Method | MNIST | | FEMNIST | | CIFAR10 | |
|---|---|---|---|---|---|---|
| | ACC | ASR | ACC | ASR | ACC | ASR |
| Euclidean distance | $97.96 \pm 0.05$ | $\mathbf{87.14 \pm 0.10}$ | $91.94 \pm 0.06$ | $\mathbf{87.39 \pm 0.08}$ | $62.38 \pm 0.07$ | $\mathbf{77.38 \pm 0.12}$ |
| Random | $97.88 \pm 0.09$ | $87.11 \pm 0.13$ | $91.89 \pm 0.08$ | $86.18 \pm 0.10$ | $59.82 \pm 0.11$ | $73.13 \pm 0.11$ |
| Projection | $97.81 \pm 0.10$ | $83.36 \pm 0.09$ | $91.99 \pm 0.12$ | $85.99 \pm 0.12$ | $60.62 \pm 0.08$ | $75.02 \pm 0.10$ |
| CS | $98.08 \pm 0.07$ | $86.17 \pm 0.10$ | $91.97 \pm 0.11$ | $86.82 \pm 0.09$ | $61.70 \pm 0.10$ | $71.01 \pm 0.09$ |
| JSD | $98.06 \pm 0.11$ | $86.37 \pm 0.08$ | $91.57 \pm 0.07$ | $85.41 \pm 0.11$ | $61.91 \pm 0.12$ | $76.96 \pm 0.13$ |
| IG | $97.89 \pm 0.05$ | $81.63 \pm 0.11$ | $90.56 \pm 0.10$ | $86.98 \pm 0.10$ | $60.07 \pm 0.09$ | $76.09 \pm 0.12$ |
| Entropy | $98.10 \pm 0.09$ | $83.70 \pm 0.12$ | $92.98 \pm 0.09$ | $85.04 \pm 0.08$ | $61.26 \pm 0.11$ | $76.61 \pm 0.09$ |

Table 7: Performance comparison of BAPFL with different flipping strategies.

| Method | AR = 10% | | AR = 20% | | AR = 30% | | AR = 40% | |
|---|---|---|---|---|---|---|---|---|
| | ACC | ASR | ACC | ASR | ACC | ASR | ACC | ASR |
| **MNIST** | | | | | | | | |
| OBF | $98.10 \pm 0.09$ | $87.10 \pm 0.12$ | $97.52 \pm 0.11$ | $87.58 \pm 0.09$ | $96.67 \pm 0.10$ | $88.18 \pm 0.11$ | $96.79 \pm 0.08$ | $89.40 \pm 0.10$ |
| GPF | $97.71 \pm 0.12$ | $77.18 \pm 0.09$ | $98.15 \pm 0.08$ | $83.23 \pm 0.10$ | $98.21 \pm 0.11$ | $85.67 \pm 0.12$ | $98.01 \pm 0.09$ | $87.57 \pm 0.13$ |
| PFS | $97.96 \pm 0.05$ | $87.14 \pm 0.10$ | $97.85 \pm 0.06$ | $\mathbf{88.38 \pm 0.12}$ | $96.89 \pm 0.07$ | $\mathbf{88.89 \pm 0.09}$ | $96.90 \pm 0.05$ | $\mathbf{91.08 \pm 0.11}$ |
| **FEMNIST** | | | | | | | | |
| OBF | $91.69 \pm 0.11$ | $83.45 \pm 0.09$ | $91.62 \pm 0.10$ | $85.36 \pm 0.12$ | $90.49 \pm 0.09$ | $86.60 \pm 0.10$ | $89.90 \pm 0.08$ | $87.56 \pm 0.11$ |
| GPF | $91.11 \pm 0.12$ | $81.88 \pm 0.11$ | $91.06 \pm 0.08$ | $83.02 \pm 0.10$ | $89.93 \pm 0.10$ | $84.23 \pm 0.12$ | $89.56 \pm 0.11$ | $85.82 \pm 0.08$ |
| PFS | $91.94 \pm 0.06$ | $87.39 \pm 0.08$ | $91.29 \pm 0.05$ | $\mathbf{88.48 \pm 0.09}$ | $90.55 \pm 0.07$ | $\mathbf{89.19 \pm 0.11}$ | $89.18 \pm 0.06$ | $\mathbf{89.23 \pm 0.10}$ |
| **CIFAR-10** | | | | | | | | |
| OBF | $62.21 \pm 0.10$ | $73.04 \pm 0.09$ | $60.23 \pm 0.08$ | $73.19 \pm 0.11$ | $58.57 \pm 0.09$ | $73.21 \pm 0.10$ | $64.67 \pm 0.12$ | $73.43 \pm 0.08$ |
| GPF | $63.54 \pm 0.11$ | $72.86 \pm 0.10$ | $61.45 \pm 0.09$ | $73.33 \pm 0.12$ | $58.36 \pm 0.10$ | $73.48 \pm 0.08$ | $61.06 \pm 0.13$ | $73.56 \pm 0.09$ |
| PFS | $62.38 \pm 0.07$ | $77.38 \pm 0.12$ | $61.47 \pm 0.08$ | $\mathbf{77.78 \pm 0.11}$ | $60.93 \pm 0.06$ | $\mathbf{78.20 \pm 0.10}$ | $60.83 \pm 0.07$ | $\mathbf{82.00 \pm 0.12}$ |

## G  COMPARISON OF DIFFERENT SAMPLE SELECTION STRATEGIES

To validate the effectiveness of our Euclidean-distance-based sample selection strategy, we replace this strategy in BAPFL with other sample selection strategies based on different measurement standards and conducted a comparison. Specifically, we further compare the our sample selection strategy against strategies based on different measures, including random selection, prototype projection (Projection), cosine similarity (CS), Jensen-Shannon divergence (JSD), information gain (IG) and prediction entropy (Entropy). For each method, malicious clients first assign an attack score to every trigger-embedded sample according to the corresponding metric, and then select the top-$K$ most "poisonous" samples to construct poisoned prototypes and optimize their local models. The implementation details of these strategies are shown in our released code (see the `get_next_poison_all_train_batch` function).

We report the average main-task accuracy (ACC) and the attack success rate (ASR) of benign clients across three datasets: MNIST, FEMNIST, and CIFAR10, under the same setting of 10% malicious clients and a $10 \times 10$ trigger. The results are summarized in Table 6.

From Table 6, we observe that the Euclidean distance consistently achieves a strong balance between preserving benign task performance and maximizing attack effectiveness. On MNIST and FEMNIST, the Euclidean distance rule attains the highest ASR while maintaining competitive ACC. On CIFAR10, the Euclidean distance rule outperforms other strategies by a clear margin in both ACC and ASR. This confirms that the sample selection strategy based on the Euclidean distance is indeed the most effective choice for PFL. This is because the prototype optimization of PFL is formulated and optimized with the Euclidean distance, Euclidean distance ensures that the sample-selection strategy is consistent with the underlying optimization mechanism of PFL. This consistency enables Euclidean distance to more accurately capture the directional and magnitude shifts introduced by triggers, thereby identifying the most poisonous samples and consistently yielding superior ASR.

## H  COMPARISON OF DIFFERENT FLIPPING STRATEGIES

To validate the effectiveness of our proposed prototype flipping strategy (PFS) in the PPS, we compare it against two intuitive baselines: 1) *Origin-based flipping (OBF)*. This strategy reflects the

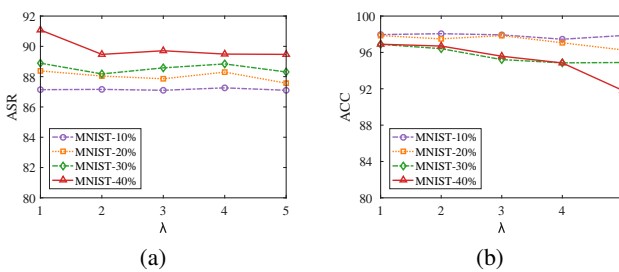

Figure 7: Effect of the weight $\lambda$ on ASR and ACC of BAPFL on the MNIST dataset.

trigger prototype $P_{tr}^{(k)}$ based on the origin to construct the poisoned prototype $P_{c*}^{(k)}$, i.e.,

$$P_{c*}^{(k)} = -P_{tr}^{(k)}. \tag{22}$$

Although simple and effective, OBF reduces the stealthiness of the attack since it often yields unstable or easily detectable poisoned prototypes that are significantly different from the benign prototypes. 2) *Global prototype-based flipping (GPF)*. This strategy reflects $P_{tr}^{(k)}$ with respect to the global prototype $\bar{P}^{(k)}$ to construct $P_{c*}^{(k)}$, i.e.,

$$P_{c*}^{(k)} = 2 \cdot \bar{P}^{(k)} - P_{tr}^{(k)}. \tag{23}$$

Compared with OBF, GPF achieves finer control over the direction of $P_{c*}^{(k)}$, but it lacks control over the norm of $P_{c*}^{(k)}$, potentially weakening attack effectiveness or introducing excessive perturbation.

Conversely, our PFS enables fine-grained control over both the direction and norm of $P_{c*}^{(k)}$, achieving more precise manipulation of the global prototype while preserving stealth.

**Experimental Comparison.** In PFL, we report the main task accuracy (ACC) and ASR of BAPFL with different flipping strategies across three datasets (MNIST, FEMNIST, CIFAR-10) and varying attack rates (AR = 10% to 40%). The results are shown in Table 7. Across all datasets and attack rates, the BAPFL with PFS consistently achieves the highest ASR while maintaining comparable or even better ACC than other baselines. This demonstrates that our strategy provides a more effective and stealthy attack mechanism by precisely constructing the direction and norm of the poisoned prototypes.

## I  THE SENSITIVITY ANALYSIS FOR ADDITIONAL HYPERPARAMETERS

### I.1  THE EFFECT OF $\lambda$ FOR BAPFL

To assess the effect of the weight $\lambda$ of $\mathcal{L}_P$ for BAPFL, we evaluate the ASR and ACC of BAPFL in PFL with $\lambda \in \{1, 2, 3, 4, 5\}$. Figure 7 presents the experimental results on the MNIST dataset. We observe that BAPFL achieves consistently high ASR across all settings, with only minor fluctuations across different $\lambda$ values. Moreover, the ACC of BAPFL generally remains stable. However, under high attack rates, the ACC of BAPFL decreases slightly as $\lambda$ increases. For example, when the attack rate is 40%, the ACC value drops from 97% ($\lambda = 1$) to 92% ($\lambda = 5$). The attacker can mitigate this effect by reducing the attack rate. Overall, BAPFL demonstrates its effectiveness under different $\lambda$ settings.

### I.2  ANALYSIS OF $\alpha$

According to Eq. equation 5, the hyperparameter $\alpha$ controls the relative importance of the backdoor task compared to the main classification task. When $\alpha$ is close to 0, the optimization is dominated by the clean task, which preserves high ACC but fails to inject an effective backdoor, resulting in a low ASR. In contrast, when $\alpha = 1$, the compromised client fully ignores the clean task and

Table 8: Effect of $\alpha$ on ACC and ASR across three datasets, AR = 10%.

| $\alpha$ | MNIST | | FEMNIST | | CIFAR-10 | |
|---|---|---|---|---|---|---|
| | ACC | ASR | ACC | ASR | ACC | ASR |
| 1 | 58.40±0.08 | 29.70±0.12 | 66.20±0.10 | 25.10±0.11 | 33.50±0.09 | 30.60±0.10 |
| 0.75 | 97.96±0.05 | **87.14±0.10** | 91.94±0.06 | **87.39±0.08** | 62.38±0.07 | **77.38±0.12** |
| 0.5 | 97.54±0.07 | 53.49±0.11 | 90.60±0.09 | 25.50±0.12 | 64.20±0.10 | 48.90±0.13 |
| 0.25 | 97.90±0.12 | 29.20±0.08 | 91.40±0.05 | 14.90±0.09 | 64.70±0.11 | 22.50±0.08 |

only optimizes for the backdoor objective. This leads to a severe drop in ACC on benign clients, indicating that the injected model is no longer useful for the original task.

We conducted additional experiments on MNIST, FEMNIST, and CIFAR-10 to systematically evaluate the effect of $\alpha$, with results reported in Table 8. The results demonstrate that setting $\alpha = 0.75$ achieves the best trade-off: the backdoor is injected with high effectiveness (high ASR) while maintaining competitive benign performance (high ACC). Extremely large or small values of $\alpha$ disrupt this balance, either by weakening the backdoor effect or by severely damaging clean accuracy.

### I.3 ANALYSIS OF $\lambda_1, \lambda_2, \lambda_3$

The three coefficients $\lambda_1, \lambda_2, \lambda_3$ correspond to the auxiliary loss terms introduced in Eq. 8. Specifically, $\lambda_1$ balances the prototype alignment objective, while $\lambda_2$ and $\lambda_3$ regulate the stealthiness penalty by constraining the mask $M_{y_t}$ and the trigger pattern $\delta_{y_t}$, respectively.

In practice, we first rescaled each loss item to ensure that all terms are of comparable magnitude, thereby preventing any single component from dominating the optimization. After normalization, we empirically set $\lambda_1 = 0.1, \lambda_2 = 0.01, \lambda_3 = 0.001$, which results in balanced contributions across different objectives and leads to stable convergence.

We further conducted experiments with small perturbations (±50%) to $\lambda_1$, $\lambda_2$, and $\lambda_3$. As shown in Table 9, even with these perturbations, the performance (both ACC and ASR) of BAPFL remains largely unaffected, indicating that BAPFL is relatively insensitive to these hyperparameters once proper normalization is applied. This robustness highlights that the success of BAPFL does not rely on fine-tuning these parameters.

Table 9: Sensitivity analysis of $\lambda_1$, $\lambda_2$, and $\lambda_3$ for BAPFL on MNIST.

| $\lambda_1$ | $\lambda_2$ | $\lambda_3$ | AR = 10% | | AR = 20% | | AR = 30% | | AR = 40% | |
|---|---|---|---|---|---|---|---|---|---|---|
| | | | ACC | ASR | ACC | ASR | ACC | ASR | ACC | ASR |
| 0.1 | 0.01 | 0.001 | 97.96 | **87.14** | 97.85 | **88.38** | 96.89 | **88.89** | 96.90 | **91.08** |
| *0.05* | 0.01 | 0.001 | 97.93 | 86.35 | 97.82 | 87.10 | 96.84 | 87.22 | 96.78 | 89.49 |
| *0.15* | 0.01 | 0.001 | 97.90 | 87.12 | 97.78 | 88.08 | 96.85 | 88.25 | 96.81 | 90.85 |
| 0.1 | *0.005* | 0.001 | 97.91 | 86.42 | 97.79 | 87.85 | 96.87 | 88.18 | 96.83 | 90.70 |
| 0.1 | *0.015* | 0.001 | 97.94 | 86.95 | 97.83 | 87.65 | 96.83 | 87.86 | 96.79 | 89.78 |
| 0.1 | 0.01 | *0.0005* | 97.92 | 86.05 | 97.81 | 87.95 | 96.83 | 87.40 | 96.79 | 88.89 |
| 0.1 | 0.01 | *0.0015* | 97.89 | 86.48 | 97.76 | 88.05 | 96.82 | 88.12 | 96.80 | 89.67 |

## J CASE STUDY

To further illustrate the effectiveness of the PPS and TOM of BAPFL, we conduct a case study on FEMNIST, in which we visualize the prototype distribution and the classification results of trigger prototypes for a benign client under three attack strategies: BAPFL(DBA), BAPFL(DBA+PPS), and BAPFL(PPS+TOM). As shown in Figure 8(a), the trigger prototypes of BAPFL(DBA) are optimized toward the global prototype of the target label, but most of them are still close to their corresponding benign prototypes. This leads to a low ASR. In contrast, in the case of BAPFL(DBA+PPS) (Figure 8(b)), PPS separates the benign prototypes from the trigger prototypes. This enables more trigger prototypes to approach the global prototype of the target label, which increases the ASR. Finally, in

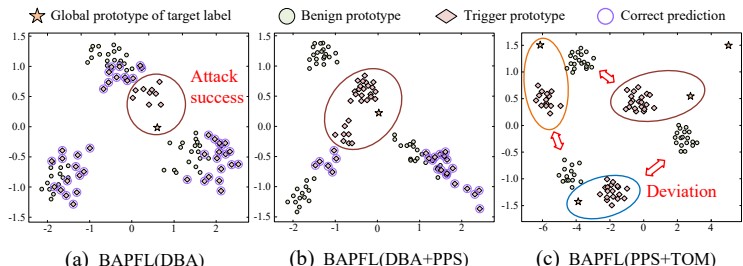

Figure 8: The principal component analysis (PCA) visualization of benign and trigger prototypes for benign client 1 under different attack strategies. The trigger prototypes classified as the original label are marked with purple circles.

Table 10: Training time comparison between benign training and BAPFL-based malicious training.

| Dataset | Benign Training | Malicious Training |
|---------|-----------------|--------------------|
| MNIST | 139.55 s | 168.25 s |
| FEMNIST | 490.34 s | 562.05 s |
| CIFAR-10 | 445.84 s | 510.46 s |
| CIFAR-100 | 585.23 s | 744.78 s |

the case of BAPFL(PPS+TOM) (Figure 8(c)), TOM further expands the target label space and enhances the alignment of trigger prototypes with the global prototype of the target labels, achieving the highest ASR across all attack strategies. The above results indicate that both PPS and TOM in BAPFL play a crucial role in enhancing the ASR.

## K    COMPUTATION AND COMMUNICATION OVERHEAD ANALYSIS OF BAPFL

**Computation Overhead**. Compared with benign training, BAPFL-based malicious training incurs additional computation due to the trigger optimization mechanism and prototype poisoning strategy. To evaluate this overhead, we measure the training time of client 1 over 200 rounds of benign training and malicious training. The results are summarized in Table 10. From Table 10, we can see that the overhead of malicious training is only slightly higher than that of benign training. Specifically, the additional cost introduced by malicious training is only 20% for MNIST, 15% for FEMNIST, 14% for CIFAR-10, and 27% for CIFAR-100. This is because the time-consuming trigger optimization in BAPFL only needs to be performed for three rounds (each with 50 epochs) to achieve convergence, and no further trigger optimization is required afterward. Such a training setup ensures that the computation overhead introduced by the BAPFL attack remains minimal.

**Communication Overhead**. The number of local prototypes uploaded by the client 1 does not change between benign and malicious training. Only the parameter values within the local prototypes are modified. Therefore, BAPFL does not introduce additional communication overhead.

## L    PERFORMANCE OF BAPFL UNDER TYPICAL BACKDOOR ATTACK SETTINGS

In this section, we analyze the performance of BAPFL under typical settings, where the number of clients are set to 200 and 10% of clients are randomly selected to participate in each round. We conduct experiments on two benchmark datasets, MNIST and CIFAR-10, and compare the results of BAPFL with baseline methods, the details are shown in Table 11.

As shown in Table 11, BAPFL consistently outperforms all baseline methods across different attack rates in both ACC and ASR. Notably, BAPFL achieves the highest ASR at all attack rates in both datasets while maintaining competitive or superior accuracy compared to the baselines. For instance,

Table 11: ACC and ASR of BAPFL and baselines under typical backdoor attack settings.

| Defense | AR=10% | | AR=20% | | AR=30% | | AR=40% | |
|---|---|---|---|---|---|---|---|---|
| | ACC | ASR | ACC | ASR | ACC | ASR | ACC | ASR |
| **MNIST** | | | | | | | | |
| MR | 96.19 | 11.34 | 96.50 | 20.27 | 96.45 | 32.09 | 96.55 | 44.12 |
| DBA | 96.58 | 22.15 | 96.25 | 26.37 | 96.77 | 36.45 | 96.82 | 45.52 |
| PFedBA | 97.25 | 18.92 | 96.85 | 24.23 | 96.36 | 37.77 | 96.55 | 47.58 |
| BapFL | 97.54 | 25.16 | 97.34 | 30.27 | 96.95 | 39.65 | 96.86 | 48.15 |
| Bad-PFL | 97.25 | 22.51 | 96.65 | 27.60 | 96.45 | 34.85 | 96.55 | 43.14 |
| Chameleon | 96.85 | 26.35 | 97.17 | 31.55 | 96.82 | 41.16 | 96.90 | 46.20 |
| A3FL | 97.35 | 28.45 | 97.20 | 34.15 | 96.85 | 42.65 | 97.05 | 47.05 |
| BAPFL | 97.82 | **71.34** | 97.88 | **75.12** | 96.87 | **80.12** | 96.96 | **82.51** |
| **CIFAR-10** | | | | | | | | |
| MR | 57.15 | 10.11 | 57.00 | 13.21 | 55.43 | 14.43 | 55.16 | 14.61 |
| DBA | 58.38 | 9.23 | 57.44 | 12.33 | 57.85 | 13.53 | 58.84 | 14.34 |
| PFedBA | 57.76 | 10.34 | 58.49 | 13.13 | 57.98 | 15.25 | 57.78 | 16.24 |
| BapFL | 58.18 | 10.32 | 59.49 | 14.32 | 58.75 | 18.53 | 57.56 | 20.22 |
| Bad-PFL | 59.54 | 10.22 | 58.98 | 15.13 | 59.54 | 16.34 | 57.41 | 18.62 |
| Chameleon | 57.55 | 8.32 | 58.60 | 11.32 | 58.11 | 14.22 | 59.39 | 15.23 |
| A3FL | 62.35 | 12.33 | 62.18 | 16.53 | 62.05 | 17.83 | 61.92 | 19.53 |
| BAPFL | 62.18 | **73.82** | 62.05 | **73.92** | 61.25 | **75.42** | 61.12 | **77.82** |

in MNIST, BAPFL achieves the highest ASR values (ranging from 71.34% to 82.51%), significantly surpassing other methods such as MR, DBA, and Chameleon, while still preserving high accuracy. These results highlight BAPFL's robustness and effectiveness in typical backdoor attack settings.

