# OpenReview forum: "BAPFL: Exploring Backdoor Attacks Against Prototype-based Federated Learning"
_ICLR.cc/2026/Conference — Submitted to ICLR 2026_

### Official Review · Reviewer_EUSX · 2025-10-31

**Soundness:** 3
**Presentation:** 3
**Contribution:** 3
**Rating:** 6
**Confidence:** 3

**Summary:**

This paper investigates the backdoor vulnerability of Prototypical Federated Learning (PFL). This paper first analyzes PFL's natural resistance to traditional attacks, then proposes BAPFL, the first backdoor attack specifically designed for PFL. BAPFL combines a Prototype Poisoning Strategy (PPS) to manipulate global prototypes via prototype flipping, and a Trigger Optimization Mechanism (TOM) to learn stealthy triggers aligned with target-class prototypes. Experiments show BAPFL achieves a high ASR while maintaining ACC.

**Strengths:**

1.This paper is the first to systematically study the backdoor security issues in PFL, an important variant of FL, filling a research gap.

2.The paper fully validates the effectiveness of BAPFL's components and its robustness under various settings through extensive internal comparative experiments.

3.This paper is well written and easy to follow.

**Weaknesses:**

1. Fails to evaluate BAPFL against SOTA general-purpose backdoor defenses (e.g., FLAME [1]), leading to an insufficient assessment of its actual threat.

2.  Lacks evaluation against more complex, PFL-specific backdoor defense strategies, failing to demonstrate robustness against such targeted defenses.

3.  Lacks comparison against other SOTA attacks (e.g., Chameleon [2], A3FL [3]).   This is needed to benchmark BAPFL's relative strength, even if those attacks require adaptation to PFL.

4. The stealthiness claim relies solely on ACC, overlooking statistical anomaly analysis of the poisoned prototypes themselves.

Reference:

[1] Nguyen et al. FLAME: Taming backdoors in federated learning.USENIX Security’ 2022.

[2] Yanbo Dai, Songze Li. Chameleon: Adapting to Peer Images for Planting Durable Backdoors in Federated Learning. ICML2023.

[3] Zhang, Hangfan, et al. A3FL: adversarially adaptive backdoor attacks to federated learning. NeurIPS’ 2023.

**Questions:**

Please refer to the weaknesses.

---

> ### Author Response · Authors · 2025-11-23
>
> **Response to W1**
>
> *"Fails to evaluate BAPFL against SOTA general-purpose backdoor defenses (e.g., FLAME)."*
>
> Thank you for your valuable comment. To address this concern, we have evaluated BAPFL against SOTA general-purpose backdoor defenses, including FLAME and Deepsight [1]. We have implemented FLAME and Deepsight in the PFL framework and evaluated BAPFL's robustness against these defenses. The experimental results (see Table below) demonstrate that BAPFL can effectively evade both FLAME and Deepsight, achieving at least 60% ASR. This is because BAPFL utilizes the prototype poisoning strategy to precisely generate the poisoned prototypes, which remain statistically similar to benign prototypes, making them difficult to detect by existing defense mechanisms.
>
> |Method|AR=10%|AR=10%|AR=20%|AR=20%|AR=30%|AR=30%|AR=40%|AR=40%|
> |---|:---:|:---:|:---:|:---:|:---:|:---:|:---:|:---:|
> ||ACC|ASR|ACC|ASR|ACC|ASR|ACC|ASR|
> |||||MNIST|||||
> |FLAME|98.01%|63.58%|98.06%|65.48%|97.82%|69.84%|98.28%|70.21%|
> |Deepsight|97.91%|64.63%|97.87%|66.23%|98.19%|71.21%|98.37%|72.99%|
> |||||FEMNIST|||||
> |FLAME|90.12%|61.60%|89.89%|63.51%|89.67%|67.13%|89.92%|71.41%|
> |Deepsight|90.87%|62.71%|90.77%|65.40%|90.82%|70.27%|91.15%|72.40%|
> |||||CIFAR-10|||||
> |FLAME|57.55%|71.36%|58.60%|73.42%|58.11%|75.95%|57.41%|77.21%|
> |Deepsight|59.54%|72.60%|58.98%|74.63%|59.50%|76.21%|59.39%|77.58%|
>
> [1] Rieger, Phillip, et al. "Deepsight: Mitigating backdoor attacks in federated learning through deep model inspection." arXiv preprint, 2022.
>
> **Response to W2**
>
> *"Lacks evaluation against more complex, PFL-specific backdoor defense strategies."*
>
> Thank you for pointing this out. To the best of our knowledge, the proposed BAPFL is the first backdoor attack method designed for prototype-based federated learning (PFL). As such, PFL-specific backdoor defense strategies have not yet been extensively explored. Therefore, in this paper, we primarily evaluate the robustness of BAPFL using existing mainstream FL-specific backdoor defense strategies.
>
> **Response to W3**
>
> *"Lacks comparison against other SOTA attacks (e.g., Chameleon, A3FL)."*
>
> Thank you for the thoughtful critique. To benchmark BAPFL's relative strength, we have implemented Chameleon and A3FL in the PFL framework (see Table below and Section 5.2.1 of the revised paper). The results demonstrate that our BAPFL method outperforms these SOTA methods. This is attributed to BAPFL's unique prototype poisoning strategy tailored for PFL, which misleads the optimization of the global prototype, and its multi-trigger optimization mechanism that adapts to the heterogeneous label distribution across clients.
>
> |Method|AR=10%|AR=10%|AR=20%|AR=20%|AR=30%|AR=30%|AR=40%|AR=40%|
> |---|:---:|:---:|:---:|:---:|:---:|:---:|:---:|:---:|
> ||ACC|ASR|ACC|ASR|ACC|ASR|ACC|ASR|
> |||||MNIST|||||
> |Chameleon|97.85%|32.10%|97.68%|38.25%|97.55%|44.34%|97.41%|49.75%|
> |A3FL|97.90%|34.82%|97.75%|40.90%|97.62%|47.16%|97.45%|52.60%|
> |||||FEMNIST|||||
> |Chameleon|91.82%|28.45%|91.50%|33.90%|91.25%|38.55%|90.85%|44.08%|
> |A3FL|91.90%|30.25%|91.65%|36.80%|91.33%|41.90%|90.95%|47.10%|
> |||||CIFAR-10|||||
> |Chameleon|62.46%|12.86%|61.90%|16.55%|61.75%|21.47%|61.50%|25.90%|
> |A3FL|62.35%|14.64%|61.80%|18.90%|61.60%|23.56%|61.35%|27.85%|
>
> **Response to W4**
>
> *"The stealthiness claim relies solely on ACC, overlooking statistical anomaly analysis of the poisoned prototypes themselves."*
>
> Thank you for your comment. We would like to clarify that in Section 5.2.2 of the original paper, we have already conducted a statistical anomaly analysis of the poisoned prototypes. Specifically, we attempt to filter out these poisoned prototypes using Multi-Krum [1], Median [2], Clipping [3], Sign [4], FLAME [5], and Deepsight [6]. Our experimental results demonstrate that the poisoned prototypes generated by BAPFL can effectively evade these defense strategies, since it leverages PPS to precisely control both the magnitude and direction of poisoned prototypes, rendering them indistinguishable from benign ones.
>
> [1] Peva Blanchard, et al. Machine learning with adversaries: Byzantine tolerant gradient descent. In Neural Information Processing Systems, 2017.
>
> [2] Zhuangzhuang Zhang, et al. Lsfl: A lightweight and secure federated learning scheme for edge computing. IEEE Transactions on Information Forensics and Security, 2023.
>
> [3] Hongyi Wang, et al. Attack of the tails: yes, you really can backdoor federated learning. In Proceedings of the 34th International Conference on Neural Information Processing Systems, 2020.
>
> [4] Zhenyuan Guo, et al. Fedsign: A sign-based federated learning framework with privacy and robustness guarantees. Computers Security, 2023.
>
> [5] Thien Duc Nguyen, et al. FLAME: Taming backdoors in federated learning. In 31st USENIX Security Symposium, 2022.
>
> [6] Phillip Rieger, et al. Deepsight: Mitigating backdoor attacks in federated learning through deep model inspection. In 29th Annual Network and Distributed System Security Symposium, 2022.

---

### Official Review · Reviewer_aqg4 · 2025-10-31

**Soundness:** 3
**Presentation:** 3
**Contribution:** 3
**Rating:** 4
**Confidence:** 5

**Summary:**

The paper investigates security vulnerabilities in Prototype-based Federated Learning (PFL) under backdoor attacks. The authors identify that PFL’s prototype-based learning and data heterogeneity provide inherent resistance to conventional attacks. To address this limitation, the paper introduces BAPFL, a novel attack framework combining a Prototype Poisoning Strategy (PPS)and a Trigger Optimization Mechanism (TOM). PPS manipulates global prototypes to mislead benign clients, while TOM learns stealthy triggers aligned with target prototypes. Extensive experiments on MNIST, FEMNIST, and CIFAR-10 across multiple PFL variants demonstrate that BAPFL achieves a substantial improvement in attack success rate (33%–75%) over traditional backdoor attacks while maintaining comparable main-task accuracy. The results further highlight BAPFL’s robustness against advanced defenses and its adaptability to heterogeneous data settings.

**Strengths:**

1. Presents the first systematic investigation of backdoor attacks in PFL and identifies its intrinsic resistance to conventional attacks.
2. The integration of PPS and TOM for dual-direction prototype optimization is well-motivated and significantly enhances both attack effectiveness and stealthiness.
3. Appendix I demonstrates that BAPFL introduces minimal computational and communication overhead, suggesting its practical feasibility.

**Weaknesses:**

1. While ``stealthiness'' is discussed as a key objective, the paper lacks visual examples of trigger-embedded samples. Figure 3 visualizes prototypes but does not illustrate the actual modified inputs seen by benign clients. Including such examples would strengthen the claim of imperceptibility.
2. The statement that robust aggregation defenses exhibit ``limited effectiveness'' against BAPFL would benefit from deeper analysis. A theoretical discussion clarifying *why* prototype manipulation circumvents these defenses would provide stronger insight.
3. Appendix G.3 asserts that BAPFL is relatively insensitive to hyperparameters after normalization, yet this claim relies on limited perturbation testing. Presenting quantitative results (e.g., plots or tables) for small variations of λ₁, λ₂, and λ₃ would substantiate this conclusion.
4. The federated learning setup deviates from standard backdoor attack settings, which typically involve 100–200 clients with partial participation (e.g., 10% per round). The current configuration of 20 clients with full participation is non-standard; clarification and evaluation under typical settings are necessary.
5. The related work section omits discussion of several defense strategies proposed for FL backdoor attacks, particularly outlier-detection-based methods [1, 2]. Evaluation beyond robust aggregation would provide a more comprehensive comparison.
6. Experiments on more complex datasets (e.g., CIFAR-100 or Tiny-ImageNet) would strengthen claims of generalizability across diverse data distributions.
7. The large performance gap in ASR (33%–75%) compared to ``traditional backdoor attacks'' raises concerns about baseline adaptation. The paper should clarify whether baselines were properly adjusted for the PFL context or justify why they cannot be effectively adapted.
8. Minor: The meaning of ``attack rate'' (e.g., proportion of malicious clients) should be explicitly defined in Section 5.1 for clarity.

[1]. Nguyen, Thien Duc, et al. "{FLAME}: Taming backdoors in federated learning." *31st USENIX Security Symposium (USENIX Security 22)*. 2022.

[2]. Rieger, Phillip, et al. "Deepsight: Mitigating backdoor attacks in federated learning through deep model inspection." arXiv preprint arXiv:2201.00763 (2022).

**Questions:**

Please address the aforementioned weaknesses and clarifications, particularly regarding experimental setup consistency and baseline adaptation.

---

> ### Author Response · Authors · 2025-11-23
>
> **Response to W1**
>
> *"The paper lacks visual examples of trigger-embedded samples. "*
>
> Thank you for the suggestion. To further strengthen the stealthiness claim of BAPFL, we have supplemented the visual examples of trigger-embedded samples in Appdix D in the revised paper. These examples clearly show that the triggers generated by BAPFL introduce only minimal and visually imperceptible modifications, demonstrating the high stealthiness of our attack.
>
> **Response to W2**
>
> *"A theoretical discussion clarifying why prototype manipulation circumvents these defenses would provide stronger insight."*
>
> Thank you for the valuable suggestion. To strengthen the explanation of BAPFL’s stealthiness, we have added a theoretical analysis of how BAPFL bypasses robust aggregation defenses in Appendix F of the revised paper.
>
> In this analysis, we first define the aggregation region of the robust aggregation function *F*, where outlier-based defenses reject prototypes that significantly deviate from the majority distribution. Second, we explain that in BAPFL, the trigger is optimized to be stealthy, ensuring that the trigger prototype remains close to both the benign prototype and the global prototype. As a result, the trigger prototype stays within the aggregation region of *F*. Third, we show that the distance between the poisoned prototype and the global prototype is equal to the distance between the global prototype and the trigger prototype. Therefore, the poisoned prototype also lies within the aggregation region of *F*. Since both the trigger and poisoned prototypes stay within the inlier aggregation region of *F*, they cannot be detected or removed by any similarity-based or distance-based defense strategy.
>
> **Response to W3**
>
> *"Presenting quantitative results (e.g., plots or tables) for small variations of $\lambda_1$, $\lambda_2$ and $\lambda_3$ would substantiate the conclusion in Appendix G.3."*
>
> Thank you for the suggestion. To strengthen the robustness claim of BAPFL, we have added a quantitative analysis of $\lambda_1$, $\lambda_2$ and $\lambda_3$ in Appendix I.3 in the revised paper, and the results (see Table below) further substantiate the conclusion that BAPFL is relatively insensitive to these hyperparameters.
>
> |$\lambda_1$|$\lambda_2$|$\lambda_3$|AR=10%|AR=10%|AR=20%|AR=20%|AR=30%|AR=30%|AR=40%|AR=40%|
> |---|---|---|---|---|---|---|---|---|---|---|
> ||||ACC|ASR|ACC|ASR|ACC|ASR|ACC|ASR|
> |0.1|0.01|0.001|97.96|87.14|97.85|88.38|96.89|88.89|96.90|91.08|
> |*0.05*|0.01|0.001|97.93|86.35|97.82|87.10|96.84|87.22|96.78|89.49|
> |*0.15*|0.01|0.001|97.90|87.12|97.78|88.08|96.85|88.25|96.81|90.85|
> |0.1|*0.005*|0.001|97.91|86.42|97.79|87.85|96.87|88.18|96.83|90.70|
> |0.1|*0.015*|0.001|97.94|86.95|97.83|87.65|96.83|87.86|96.79|89.78|
> |0.1|0.01|*0.0005*|97.92|86.05|97.81|87.95|96.83|87.40|96.79|88.89|
> |0.1|0.01|*0.0015*|97.89|86.48|97.76|88.05|96.82|88.12|96.80|89.67|
>
> **Response to W4**
>
> *"The federated learning setup deviates from standard backdoor attack settings, clarification and evaluation under typical settings are necessary."*
>
> Thank you for the valuable suggestion. We have conducted experiments under typical backdoor attack settings, i.e., the number of clients are set to 200 and 10% clients are selected to participate the training per round (see Table below). We have added the experimental results and analysis to Appendix L in the revised paper. The results show that BAPFL still outperforms the baseline methods, although its ASR is slightly lower compared to the results obtained under our original experimental settings. This is attributed to BAPFL's unique prototype poisoning strategy tailored for PFL, which misleads the optimization of the global prototype, and its multi-trigger optimization mechanism that adapts to the heterogeneous label distribution across clients.
>
> |Method|AR=10%|AR=10%|AR=20%|AR=20%|AR=30%|AR=30%|AR=40%|AR=40%|
> |---|:---:|:---:|:---:|:---:|:---:|:---:|:---:|:---:|
> ||ACC|ASR|ACC|ASR|ACC|ASR|ACC|ASR|
> |||||MNIST|||||
> |MR|96.19|11.34|96.50|20.27|96.45|32.09|96.55|44.12|
> |DBA|96.58|22.15|96.25|26.37|96.77|36.45|96.82|45.52|
> |PFedBA|97.25|18.92|96.85|24.23|96.36|37.77|96.55|47.58|
> |BapFL|97.54|25.16|97.34|30.27|96.95|39.65|96.86|48.15|
> |Bad-PFL|97.25|22.51|96.65|27.60|96.45|34.85|96.55|43.14|
> |Chameleon|96.85|26.35|97.17|31.55|96.82|41.16|96.90|46.20|
> |A3FL|97.35|28.45|97.20|34.15|96.85|42.65|97.05|47.05|
> |BAPFL|97.82|71.34|97.88|75.12|96.87|80.12|96.96|82.51|
> |||||CIFAR-10||||||
> |MR|57.15|10.11|57.00|13.21|55.43|14.43|55.16|14.61|
> |DBA|58.38|9.23|57.44|12.33|57.85|13.53|58.84|14.34|
> |PFedBA|57.76|10.34|58.49|13.13|57.98|15.25|57.78|16.24|
> |BapFL|58.18|10.32|59.49|14.32|58.75|18.53|57.56|20.22|
> |Bad-PFL|59.54|10.22|58.98|15.13|59.54|16.34|57.41|18.62|
> |Chameleon|57.55|8.32|58.60|11.32|58.11|14.22|59.39|15.23|
> |A3FL|62.35|12.33|62.18|16.53|62.05|17.83|61.92|19.53|
> |BAPFL|62.18|73.82|62.05|73.92|61.25|75.42|61.12|77.82|

---

> ### Author Response · Authors · 2025-11-23
>
> **Response to W5**
>
> *"The related work section omits discussion of several defense strategies proposed for FL backdoor attacks. Evaluation beyond robust aggregation would provide a more comprehensive comparison."*
>
> Thank you for this helpful comment. In our revised paper, we have supplemented the discussion of several defense strategies proposed for FL backdoor attacks, including FLAME and Deepsight, in Section 2. Additionally, in Section 5.2.2, we have implemented FLAME and Deepsight in the PFL framework and evaluated BAPFL's robustness against these defenses (see Table below). We observe that FLAME achieves the strongest defensive effects, even outperforming the defense strategies discussed in our original paper. However, even with FLAME, BAPFL still achieves at least 60% ASR while maintaining a high ACC. The key reason is that BAPFL leverages PPS to precisely control both the magnitude and direction of poisoned prototypes, rendering them indistinguishable from benign ones.
>
> |Method|AR=10%|AR=10%|AR=20%|AR=20%|AR=30%|AR=30%|AR=40%|AR=40%|
> |---|:---:|:---:|:---:|:---:|:---:|:---:|:---:|:---:|
> ||ACC|ASR|ACC|ASR|ACC|ASR|ACC|ASR|
> |||||MNIST|||||
> |FLAME|98.01%|63.58%|98.06%|65.48%|97.82%|69.84%|98.28%|70.21%|
> |Deepsight|97.91%|64.63%|97.87%|66.23%|98.19%|71.21%|98.37%|72.99%|
> |||||FEMNIST|||||
> |FLAME|90.12%|61.60%|89.89%|63.51%|89.67%|67.13%|89.92%|71.41%|
> |Deepsight|90.87%|62.71%|90.77%|65.40%|90.82%|70.27%|91.15%|72.40%|
> |||||CIFAR-10|||||
> |FLAME|57.55%|71.36%|58.60%|73.42%|58.11%|75.95%|57.41%|77.21%|
> |Deepsight|59.54%|72.60%|58.98%|74.63%|59.50%|76.21%|59.39%|77.58%|
>
> **Response to W6**
>
> *"Experiments on more complex datasets would strengthen claims of generalizability across diverse data distributions."*
>
> Thank you for pointing this out. To demonstrate broad generalizability of BAPFL, we have included new experiments on the CIFAR-100 dataset and adopted ResNet-101 as the local model (see Table below). BAPFL remains effective on these settings.
>
> |Method|AR=10%|AR=10%|AR=20%|AR=20%|AR=30%|AR=30%|AR=40%|AR=40%|
> |---|---|---|---|---|---|---|---|---|
> ||ACC|ASR|ACC|ASR|ACC|ASR|ACC|ASR|
> |MR|67.31±0.16|5.12±0.22|66.41±0.15|8.45±0.18|68.96±0.20|9.63±0.25|67.59±0.17|10.58±0.21|
> |DBA|67.10±0.18|8.56±0.20|66.80±0.17|10.42±0.21|66.65±0.19|12.15±0.23|67.45±0.18|13.82±0.22|
> |PFedBA|67.22±0.19|7.30±0.22|67.92±0.14|11.65±0.20|67.96±0.18|13.70±0.24|67.67±0.16|15.48±0.20|
> |BapFL|67.10±0.18|12.35±0.24|67.45±0.16|19.35±0.22|67.38±0.17|22.12±0.20|67.21±0.18|24.25±0.23|
> |Bad-PFL|67.20±0.17|11.25±0.19|66.99±0.15|15.10±0.21|67.50±0.16|19.56±0.23|67.80±0.17|20.12±0.19|
> |Chameleon|67.28±0.18|10.35±0.18|67.05±0.17|13.85±0.22|66.95±0.18|17.25±0.19|66.85±0.15|20.10±0.20|
> |A3FL|67.32±0.19|11.80±0.20|67.15±0.16|15.25±0.19|67.00±0.17|18.95±0.20|66.90±0.16|21.75±0.19|
> |BAPFL|67.38±0.15|75.67±0.18|68.67±0.16|76.22±0.15|67.17±0.19|77.81±0.17|68.02±0.14|79.82±0.16|
>
> **Response to W7**
>
> *"The paper should clarify whether baselines were properly adjusted for the PFL context or justify why they cannot be effectively adapted."*
>
> Thank you for the thoughtful critique. The baselines cannot be effectively adapted in PFL because:
>
> - Different Data-Sharing Mechanisms: The baselines typically embeds backdoor parameters into model parameters and directly transfer these poisoned parameters to other benign clients. However, in PFL, clients share class prototypes (i.e., outputs from the embedding layer of their local models), rather than model parameters, with the server for aggregation. This difference means that attackers cannot directly transfer the backdoor model parameters to benign clients, significantly reducing the attack's effectiveness.
>
> - Single vs. Multiple Target Labels: The baselines, such as PFedBA, typically focus on a single target label for poisoning. However, PFL involves label heterogeneity across clients, where some clients may not have training samples of the chosen target label. As discussed in Section 3.3, these clients’ local models are unable to learn the "trigger-target label" mapping, leading to low ASR.
>
> BAPFL overcomes these challenges in two key ways:
>
> - BAPFL generates specific poisoned prototypes to precisely manipulate the optimization tread of global prototypes and mislead the optimization of benign prototypes. This results in the benign prototypes diverging from the trigger prototypes so that the trigger-embedded samples are more likely to be misclassified by benign models.
>
> - BAPFL expands the target label space and optimizes triggers for each target label. This increases the probability that the target labels overlap with the local label space of benign clients, thereby enabling the benign models to inadvertently activate the “trigger-target label” mapping.
>
> **Response to W8**
>
> *"The meaning of "attack rate'' should be explicitly defined for clarity."*
>
> Thank you for pointing this out. We have explicitly added the definition of 'attack rate' (i.e., the proportion of malicious clients) in Section 5.1 for clarity.

---

> > ### Comment · Reviewer_aqg4 · 2025-11-25
> >
> > Thank you for your comprehensive and diligent responses to all my concerns, including the addition of extensive new experiments. Your efforts have significantly strengthened the paper's empirical validation, and overall presentation, leading me to raise my score.

---

### Official Review · Reviewer_b8XZ · 2025-10-31

**Soundness:** 3
**Presentation:** 3
**Contribution:** 2
**Rating:** 4
**Confidence:** 3

**Summary:**

This paper focuses on the security of Prototype-Based Federated Learning (PFL), a paradigm that addresses data heterogeneity in federated learning (FL) by aggregating class prototypes (mean feature vectors) instead of full model parameters. PFL exhibits inherent robustness against existing backdoor attacks, attributed to the limited influence of poisoned prototypes (only affecting the embedding layer of benign models) and client data heterogeneity (some clients lack target label samples, breaking trigger-target mappings). To fill the gap of PFL-specific backdoor attacks, the authors propose BAPFL. Selects high-impact trigger samples via Euclidean distance and flips trigger prototypes to manipulate global prototypes away from trigger prototypes, pushing benign prototypes to diverge from trigger prototypes. Expands the target label space to cover benign clients’ local labels, learns label-specific stealthy triggers, and aligns trigger prototypes with the global prototype of the target label.

**Strengths:**

1-It addresses a critical, underexplored problem—PFL’s security against backdoor attacks. Prior work primarily focuses on vanilla FL’s backdoor vulnerabilities, while PFL’s unique prototype aggregation mechanism creates a distinct security landscape; the authors are the first to systematically analyze PFL’s resistance and design a dedicated attack, which fills an important research gap.

2-BAPFL’s dual-direction optimization (PPS + TOM) is well-motivated and tailored to PFL’s characteristics. PPS targets the prototype aggregation process (a core of PFL) to disrupt benign prototype learning, while TOM addresses client heterogeneity to ensure attack coverage—this design effectively overcomes PFL’s inherent resistance and is more innovative than adapting vanilla FL attacks to PFL.

3-The paper provides formal theoretical analysis, including two key assumptions (prototype-based classification rules) and two theorems (proving PPS’s ability to increase misclassification probability and TOM’s role in activating trigger-target mappings), which strengthens the methodological validity.

**Weaknesses:**

1-The paper claims PFL is applied in smart healthcare and autonomous driving, but all experiments use standard image datasets (MNIST, CIFAR-10) with synthetic data heterogeneity. There is no validation on domain-specific datasets (e.g., medical imaging datasets like ChestX-ray14 for healthcare) or under real-world constraints (e.g., limited client resources, intermittent connectivity), making it hard to assess BAPFL’s practical impact.

2-The paper assumes the adversary controls multiple compromised clients and has full access to global prototypes. In practice, adversaries may only control a single client or have incomplete knowledge of global prototypes (e.g., due to partial participation in aggregation rounds). The paper does not evaluate BAPFL’s performance under weaker adversary models (e.g., 1 compromised client, partial global prototype knowledge), limiting its generalizability.

3-The experiments only run for 200 training rounds. In real PFL systems, training may last for thousands of rounds, and benign clients’ continuous updates could dilute the backdoor effect. The paper does not test BAPFL’s persistence over extended training, leaving uncertainty about its long-term effectiveness.

**Questions:**

1-The paper claims PFL is applied in smart healthcare and autonomous driving, but all experiments use standard image datasets (MNIST, CIFAR-10) with synthetic data heterogeneity. There is no validation on domain-specific datasets (e.g., medical imaging datasets like ChestX-ray14 for healthcare) or under real-world constraints (e.g., limited client resources, intermittent connectivity), making it hard to assess BAPFL’s practical impact.

2-The paper assumes the adversary controls multiple compromised clients and has full access to global prototypes. In practice, adversaries may only control a single client or have incomplete knowledge of global prototypes (e.g., due to partial participation in aggregation rounds). The paper does not evaluate BAPFL’s performance under weaker adversary models (e.g., 1 compromised client, partial global prototype knowledge), limiting its generalizability.

3-The experiments only run for 200 training rounds. In real PFL systems, training may last for thousands of rounds, and benign clients’ continuous updates could dilute the backdoor effect. The paper does not test BAPFL’s persistence over extended training, leaving uncertainty about its long-term effectiveness.

---

> ### Author Response · Authors · 2025-11-23
>
> **Response to W1**
>
> *“The paper claims PFL is applied in smart healthcare and autonomous driving, ...”*
>
> Thank you for the detailed comment. We respectfully clarify that our original manuscript does not claim PFL has been applied in smart healthcare or autonomous driving. We refer to FL, which is widely used in these fields, while PFL, though promising, is still in the research phase and has not yet been applied to these areas.
>
> *“There is no validation on domain-specific datasets (e.g., medical imaging datasets like ChestX-ray14 for healthcare) or under real-world constraints (e.g., limited client resources, intermittent connectivity), making it hard to assess BAPFL’s practical impact.”*
>
> For domain-specific datasets, we would like to clarify that our focus is on developing a general and effective backdoor attack method that can be integrated with different PFL frameworks, rather than the application of backdoor attacks to specific domains such as healthcare. Therefore, we conducted experiments on standard datasets, which is consistent with the experimental setup used by other general backdoor attack methods [1, 2]. This setup allows us to evaluate the effectiveness of BAPFL in a broader and more generalizable way.
>
> For the impact of BAPFL under real-world constraints, we have already addressed this concern in Appendix I of the original manuscript, where we evaluate the computation and communication overhead of deploying BAPFL. The results show that BAPFL just introduces low computation overhead and no additional communication overhead. These findings further demonstrate the practicality of BAPFL in real-world scenarios.
>
> [1] Lyu, Xiaoting, et al. “Lurking in the shadows: unveiling stealthy backdoor attacks against personalized federated learning.” In Proceedings of the 33rd USENIX Conference on Security Symposium, 2024.
>
> [2] Ye, Tiandi, et al. Bapfl: You can backdoor personalized federated learning. ACM Transactions on Knowledge Discovery from Data, 2024.
>
> **Response to W2**
>
> *"The paper does not evaluate BAPFL’s performance under weaker adversary models (e.g., 1 compromised client, partial global prototype knowledge), limiting its generalizability."*
>
> Thank you for the suggestion. We respectfully clarify that our threat model follows a standard setup, and the similar adversary’s capabilities and knowledge assumptions are consistent with those used in the baselines [1, 2, 3]. To address the reviewer’s concern, we have evaluated BAPFL's performance under weaker adversary models. Specifically, when the adversary only has incomplete knowledge of global prototypes, it can still obtains the full global prototypes by submitting forged local prototypes for specific classes and receiving the corresponding global prototypes. Additionally, we have tested the performance of BAPFL under the single compromised client settings (see Table below). BAPFL still achieves over 75% ASR on all three datasets. These additional experiments demonstrate that BAPFL retains effectiveness even under weaker adversary conditions.
>
> |Dataset|ACC|ASR|
> |---|---|---|
> |MNIST|97.80%|81.12%|
> |FMNIST|90.93%|82.12%|
> |CIFAR10|93.62%|76.71%|
>
> [1] Lyu, Xiaoting, et al. “Lurking in the shadows: unveiling stealthy backdoor attacks against personalized federated learning.” In Proceedings of the 33rd USENIX Conference on Security Symposium, 2024.
>
> [2] Ye, Tiandi, et al. Bapfl: You can backdoor personalized federated learning. ACM Transactions on Knowledge Discovery from Data, 2024.
>
> [3] Fan, Mingyuan, et al. Bad-pfl: Exploring backdoor attacks against personalized federated learning. In Proceedings of the International Conference on Learning Representations, 2025.
>
> **Response to W3**
>
> *"The experiments only run for 200 training rounds. The paper does not test BAPFL’s persistence over extended training"*
>
> We appreciate the reviewer’s concern. In all the related PFL papers, such as FedProto [1] and FedPD [2], the training rounds are also set to 200 on standard datasets, which is sufficient for model convergence.
>
> To demonstrate broad effectiveness of BAPFL, we have included new experiments with 1,000 training rounds across multiple datasets (see Table below). The results demonstrate that BAPFL’s attack effectiveness remains unaffected under this setting.
>
> |Method|AR=10%|AR=10%|AR=20%|AR=20%|AR=30%|AR=30%|AR=40%|AR=40%|
> |---|---|---|---|---|---|---|---|---|
> ||ACC|ASR|ACC|ASR|ACC|ASR|ACC|ASR|
> |MNIST  |98.13%   |87.11%   |97.85%   |88.16%   |97.17%   |88.39%   |97.26%   |91.94%   |
> |FMNIST |91.07%   |87.42%   |90.74%   |88.08%   |90.70%   |88.54%   |90.95%   |89.56%   |
> |CIFAR10|62.42%   |77.49%   |61.54%   |77.81%   |61.10%   |78.19%   |60.84%   |81.27%   |
>
> [1] Tan, Yue, et al. "Fedproto: Federated prototype learning across heterogeneous clients." Proceedings of the AAAI conference on artificial intelligence, 2022.
>
> [2] Tan, Zhou, et al. "FedPD: Defending federated prototype learning against backdoor attacks." Neural Networks, 2025.

---

### Official Review · Reviewer_CdtN · 2025-11-01

**Soundness:** 3
**Presentation:** 3
**Contribution:** 2
**Rating:** 4
**Confidence:** 4

**Summary:**

This paper identifies that Prototype-based Federated Learning (PFL) is resistant to existing backdoor attacks due to its prototype mechanism and data heterogeneity. To overcome this, the authors propose BAPFL, which integrates a Prototype Poisoning Strategy (PPS) and a Trigger Optimization Mechanism (TOM). Experimental results show that BAPFL is effective.

**Strengths:**

* The paper is well-structured.

* This is the first backdoor attack method designed for Prototype-based Federated Learning (PFL).

* The experimental results validate the attack performance of the proposed method.

**Weaknesses:**

* The experimental settings are rather unusual. The clients use a very small local batch size of 4. What does the attack rate (AR) mean? Is it the proportion of malicious clients? Based on Table 1, the proposed attack method appears to reduce the model's main task accuracy (ACC). The datasets and models used are also simplistic. The authors should report the results on large-scale datasets and models for a more convincing evaluation.

* A sensitivity analysis for $\lambda$ is missing.

* Some findings presented in Section 3.3 appear to overlap with findings in Bad-PFL.

* The paper lacks a discussion about potential defense mechanisms.

**Questions:**

See Weaknesses.

---

> ### Author Response · Authors · 2025-11-23
>
> **Response to W1**
>
> *“The experimental settings are rather unusual. The clients use a very small local batch size of 4.”*
>
> We appreciate the reviewer’s thoughtful concern. We would like to clarify that our experimental settings are consistent with those used in FedProto [1], which first proposed the prototype-based federated learning (PFL) approach. In fact, the effectiveness of the BAPFL attack does not rely on the specifics of these parameter adjustments. Because when BAPFL is deployed in different PFL frameworks such as FedPCL [2] and FedPD [3], where the local batch size is actually set to 32 and 64 respectively, BAPFL still achieves an ASR of 65%-81%.
>
> *“What does the attack rate (AR) mean? Is it the proportion of malicious clients? ”*
>
> The attack rate (AR) refers to the proportion of malicious clients. We have explicitly added the definition of 'attack rate' in Section 5.1 in the revised paper for clarity.
>
> *“Based on Table 1, the proposed attack method appears to reduce the model's main task accuracy (ACC).”*
>
> We have provided additional experimental results to demonstrate that the impact of BAPFL on the main task accuracy (ACC) is negligible. Specifically, we tested the ACC of the model under the condition of no attack, the results are shown in the Table below. We observe that, compared to the ACC results at an attack rate of 0%, the impact of BAPFL on the model's ACC is minimal when the attack rate increases to 40%. Specifically, for the MNIST dataset, the decrease in ACC is less than 1%, for the FEMNIST dataset, the decrease in ACC is less than 2.5%, and for the CIFAR-10 dataset, the decrease in ACC is less than 2%. We consider these fluctuations are generally small and negligible.
>
> |Dataset|AR=0%|AR=10%|AR=20%|AR=30%|AR=40%|
> |---|---|---|---|---|---|
> ||ACC|ACC|ACC|ACC|ACC|
> |MNIST|97.64%|97.96%|97.85%|96.89%|96.90%|
> |FMNIST|91.67%|91.94%|91.29%|90.55%|89.18%|
> |CIFAR10|62.60%|62.38%|61.47%|60.93%|60.83%|
>
>  *“The datasets and models used are also simplistic. The authors should report the results on large-scale datasets and models for a more convincing evaluation.”*
>
> We would like to clarify that the datasets and models used in our current experiments are standard and also adopted by the baseline methods [4, 5]. To demonstrate broad effectiveness of BAPFL, we have included new experiments on the CIFAR-100 dataset and adopted ResNet-101 as the local model (see Table below). BAPFL remains effective on these settings.
>
> |Method|AR=10%|AR=10%|AR=20%|AR=20%|AR=30%|AR=30%|AR=40%|AR=40%|
> |---|---|---|---|---|---|---|---|---|
> ||ACC|ASR|ACC|ASR|ACC|ASR|ACC|ASR|
> |MR|67.31±0.16|5.12±0.22|66.41±0.15|8.45±0.18|68.96±0.20|9.63±0.25|67.59±0.17|10.58±0.21|
> |DBA|67.10±0.18|8.56±0.20|66.80±0.17|10.42±0.21|66.65±0.19|12.15±0.23|67.45±0.18|13.82±0.22|
> |PFedBA|67.22±0.19|7.30±0.22|67.92±0.14|11.65±0.20|67.96±0.18|13.70±0.24|67.67±0.16|15.48±0.20|
> |BapFL|67.10±0.18|12.35±0.24|67.45±0.16|19.35±0.22|67.38±0.17|22.12±0.20|67.21±0.18|24.25±0.23|
> |Bad-PFL|67.20±0.17|11.25±0.19|66.99±0.15|15.10±0.21|67.50±0.16|19.56±0.23|67.80±0.17|20.12±0.19|
> |Chameleon|67.28±0.18|10.35±0.18|67.05±0.17|13.85±0.22|66.95±0.18|17.25±0.19|66.85±0.15|20.10±0.20|
> |A3FL|67.32±0.19|11.80±0.20|67.15±0.16|15.25±0.19|67.00±0.17|18.95±0.20|66.90±0.16|21.75±0.19|
> |BAPFL|67.38±0.15|75.67±0.18|68.67±0.16|76.22±0.15|67.17±0.19|77.81±0.17|68.02±0.14|79.82 ± 0.16|
>
> [1] Tan, Yue, et al. "Fedproto: Federated prototype learning across heterogeneous clients." Proceedings of the AAAI conference on artificial intelligence. Vol. 36. No. 8. 2022.
>
> [2] Tan, Yue, et al. "Federated learning from pre-trained models: A contrastive learning approach." Advances in neural information processing systems 35 (2022): 19332-19344.
>
> [3] Tan, Zhou, et al. "FedPD: Defending federated prototype learning against backdoor attacks." Neural Networks 184 (2025): 107016.
>
> [4] Lyu, Xiaoting, et al. “Lurking in the shadows: unveiling stealthy backdoor attacks against personalized federated learning.” In Proceedings of the 33rd USENIX Conference on Security Symposium, pp. 18, 2024.
>
> [5] Ye, Tiandi, et al. Bapfl: You can backdoor personalized federated learning. ACM Transactions on Knowledge Discovery from Data, 18(7):166, 2024.
>
> **Response to W2**
>
> *“A sensitivity analysis for $\lambda$ is missing.”*
>
> We thank the reviewer for highlighting this important concern. We would like to clarify that a sensitivity analysis for $\lambda$ has already been included in Appendix G.1 of the original manuscript. The analysis demonstrates that BAPFL achieves consistently high ACC and ASR across varying $\lambda$ values, with only minor fluctuations.

---

> > ### Author Response · Authors · 2025-11-23
> >
> > **Response to W3**
> >
> > *“Some findings presented in Section 3.3 appear to overlap with findings in Bad-PFL.”*
> >
> > We respectfully disagree with this point. The findings/challenges presented in Section 3.3 of our paper are distinct from those identified in Bad-PFL. First, the research backgrounds of BAPFL and Bad-PFL differ significantly. Our study focuses on prototype-based federated learning [1], where clients share class prototypes (i.e., outputs from the embedding layer of the local model) with the server for aggregation. In contrast, Bad-PFL is focuses on personalized federated learning, where clients share parts or full of their local model parameters for aggregation. Due to these differences in data sharing and training paradigms, the attack challenges they identified are inherently different. In BAPFL, our challenge lies in constructing poisoned prototypes that mislead the global prototype, thereby indirectly misguiding the optimization of benign prototypes. However, Bad-PFL embeds backdoor parameters into malicious model parameters, which are then directly transferred to global model and benign clients’ local models (see Algorithm 1 in Bad-PFL). Therefore, the findings/challenges between our BAPFL and Bad-PFL are fundamentally different.
> >
> > **Response to W4**
> >
> > *“The paper lacks a discussion about potential defense mechanisms.”*
> >
> > We appreciate the reviewer’s concern. We would like to clarify that our focus is primarily on the backdoor attack method in PFL, and discussing potential defense mechanisms is not an essential part of our work, as the baselines, such as Bad-PFL [1] and PFedBA [2], do not include such a discussion. However, to address the reviewer’s concern, we have already included a discussion of potential defense strategies for BAPFL in Section 5.2.2 in the revised paper. The details are also shown below:
> >
> > “..., we believe that specialized defense mechanisms are needed to effectively counter BAPFL. For instance, the server can maintain the historical information of class prototypes uploaded by each client, and visualize the optimization paths rather than the current round of these prototypes to identify poisoned prototypes. This is because the single-round update of poisoned prototypes is minimal and difficult to distinguish from the update of benign prototypes, while the optimization paths of benign and poisoned prototypes are clearly different. Specifically, in PFL, the poisoned prototypes share a consistent optimization objective, which aims to pull the global prototype away from the trigger prototype, rather than move towards the global prototype. In contrast, the benign prototypes aims to continually move towards the global prototype. Additionally, benign clients can further mitigate the effects of BAPFL by fine-tuning their local models on clean datasets and correcting the misled benign prototypes.”
> >
> > [1] Fan, Mingyuan, et al. Bad-pfl: Exploring backdoor attacks against personalized federated learning. In Proceedings of the International Conference on Learning Representations, 2025.
> >
> > [2] Lyu, Xiaoting, et al. “Lurking in the shadows: unveiling stealthy backdoor attacks against personalized federated learning.” In Proceedings of the 33rd USENIX Conference on Security Symposium, pp. 18, 2024.

---

### Author Response · Authors · 2025-11-29
**Rebuttal Summary Comment**

We sincerely thank the reviewers for their thoughtful and constructive feedback. The rebuttal process has significantly strengthened our paper. We have addressed all concerns by providing extensive new theoretical analysis, comprehensive additional experiments, and detailed clarifications, leading to a substantial improvement in the paper's overall quality and empirical validation.

**Core Improvements and Responses**

We summarize the key enhancements made in the revised manuscript:

1. **Enhanced Empirical Validation on Large-Scale Settings (W1 from R1, W3 from R2, W6 from R3):**

- We reported new results on the more complex CIFAR-100 dataset using ResNet-101 as the local model, demonstrating BAPFL's continued superior effectiveness (ASR up to 79.82% at AR = 40%).

- We showed BAPFL's persistence over an extended training period of 1,000 rounds, confirming its long-term effectiveness (ASR up to 91.94% on MNIST).

2. **Increased Robustness and Generalizability Evaluation (W2 from R2, W4 and W5 from R3, W1 and W3 from R4):**

- Against State-of-the-Art (SOTA) Defenses: We implemented and evaluated BAPFL against SOTA general-purpose backdoor defenses, FLAME [1] and Deepsight [2], demonstrating that BAPFL remains highly effective by achieving ASRs of 60%-77% across datasets. This robustness is attributed to BAPFL's precise Prototype Poisoning Strategy (PPS).

- Against SOTA Attacks: We included comparative experiments against other SOTA attacks, Chameleon [3] and A3FL [4], showing BAPFL's superior performance, which validates the necessity and effectiveness of our PFL-specific prototype poisoning approach.

- Under Weaker Adversary Models: We validated BAPFL's performance under a single compromised client setting, where it still achieves over 75% ASR on all three tested datasets.

- Under Standard FL Settings: We conducted experiments under the standard configuration of 200 clients with 10% partial participation, where BAPFL still significantly outperforms baselines.

3. **Strengthened Theoretical and Visual Analysis (W1, W2, W3 from R3, W4 from R4):**

- Stealthiness Analysis: We provided a theoretical analysis in the Appendix to clarify precisely why BAPFL's stealthy prototypes circumvent robust aggregation defenses like Multi-Krum and Median.

- Visual Examples: We added visual examples of the trigger-embedded samples, confirming that the modifications generated by BAPFL are minimal and visually imperceptible, thus strengthening the stealthiness claim.

- Hyperparameter Sensitivity: We added a quantitative sensitivity analysis for $\lambda_1, \lambda_2, \lambda_3$ using a detailed table, substantiating the claim that BAPFL is relatively insensitive to these hyperparameters.

4. **Crucial Clarifications (W1, W4 from R1, W8 from R3):**

- We confirmed that the attack rate (AR) is the proportion of malicious clients.

- We clarified the minimal impact of BAPFL on the main task accuracy (ACC), with decreases of less than 2.5% across all datasets.

- We included a discussion of potential defense mechanisms against BAPFL in the revised paper, outlining a strategy based on visualizing the optimization paths of prototypes.

**Reviewer Consensus and Score Update**

We are pleased to note that Reviewer 3 explicitly acknowledged the positive impact of our rebuttal, stating:

*"Thank you for your comprehensive and diligent responses to all my concerns, including the addition of extensive new experiments. Your efforts have significantly strengthened the paper's empirical validation, and overall presentation, leading me to raise my score."*

This positive change in assessment confirms that our additions and clarifications have successfully addressed the major concerns raised. The comprehensive, new experiments across various metrics, datasets, models, and adversary settings, combined with the new theoretical insights, demonstrate the robustness and practical threat posed by BAPFL.


[1] Nguyen, Thien Duc, et al. "FLAME: Taming backdoors in federated learning." 31st USENIX Security Symposium (USENIX Security 22). 2022.

[2] Rieger, Phillip, et al. "Deepsight: Mitigating backdoor attacks in federated learning through deep model inspection." arXiv preprint arXiv:2201.00763 (2022).

[3] Yanbo Dai, Songze Li. Chameleon: Adapting to Peer Images for Planting Durable Backdoors in Federated Learning. ICML2023.

[4] Zhang, Hangfan, et al. A3FL: adversarially adaptive backdoor attacks to federated learning. NeurIPS’ 2023.

---

### Meta-Review · Area_Chair_i8ND · 2026-01-06

**Summary:**

The paper introduces BAPFL, a backdoor attack tailored for prototype-based federated learning (PFL). While the idea addresses an underexplored security issue and includes extensive experiments, most reviewers raised concerns about practical relevance, experimental setup, and generalizability. Key issues include reliance on synthetic datasets, non-standard configurations, and assumptions about adversary capabilities. Despite improvements in the rebuttal, the overall contribution is seen as incremental and lacking strong real-world validation.

**Reviewer Concerns:**

The rebuttal added experiments on CIFAR-100, extended training rounds, and evaluations under weaker adversary models, partially addressing concerns about scalability and robustness. However, major issues remain: lack of domain-specific datasets, limited discussion of practical impact, and assumptions that may not hold in real-world PFL scenarios. These concerns were highlighted by multiple reviewers and are still outstanding.

**Reviewer Scores:**

- Reviewer CdtN (Marginally below): Likely unchanged; rebuttal added clarifications and CIFAR-100 results, but concerns about unusual settings and limited baselines persist.
- Reviewer b8XZ (Marginally below): Possibly unchanged; new experiments on weaker adversary models and extended rounds help, but practical relevance and domain-specific validation remain unresolved.
- Reviewer aqg4 (Marginally below → express intentional to raise): Score will likely increase after rebuttal if a full discussion happened;  acknowledged comprehensive new experiments and theoretical analysis, but still noted non-standard setup and missing broader comparisons.
- Reviewer EUSX (Marginally above): Might remain slightly positive; rebuttal addressed missing comparisons and defense evaluations, but stealthiness and anomaly analysis concerns are only partially mitigated.

---

### Decision · Program_Chairs · 2026-01-26

Reject